# Thioesters provide a plausible prebiotic path to proto-peptides

Moran Frenkel-Pinter[1,2,3], Marcos Bouza [1,2], Facundo M. Fernández [1,2], Luke J. Leman [1,4], Loren Dean Williams [1,2], Nicholas V. Hud [1,2✉] & Aikomari Guzman-Martinez [1,5✉]

It is widely assumed that the condensation of building blocks into oligomers and polymers was important in the origins of life. High activation energies, unfavorable thermodynamics and side reactions are bottlenecks for abiotic peptide formation. All abiotic reactions reported thus far for peptide bond formation via thioester intermediates have relied on high energy molecules, which usually suffer from short half-life in aqueous conditions and therefore require constant replenishment. Here we report plausible prebiotic reactions of mercaptoacids with amino acids that result in the formation of thiodepsipeptides, which contain both peptide and thioester bonds. Thiodepsipeptide formation was achieved under a wide range of pH and temperature by simply drying and heating mercaptoacids with amino acids. Our results offer a robust one-pot prebiotically-plausible pathway for proto-peptide formation. These results support the hypothesis that thiodepsipeptides and thiol-terminated peptides formed readily on prebiotic Earth and were possible contributors to early chemical evolution.

---

[1] NSF-NASA Center for Chemical Evolution, Georgia Institute of Technology, Atlanta, GA 30332, USA. [2] School of Chemistry & Biochemistry, Georgia Institute of Technology, Atlanta, GA 30332, USA. [3] Institute of Chemistry, The Hebrew University of Jerusalem, Jerusalem 91904, Israel. [4] Department of Chemistry, The Scripps Research Institute, La Jolla, CA 92037, USA. [5] Department of Chemistry, University of Puerto Rico, Mayagüez, Mayagüez, PR 00681, USA. ✉email: hud@chemistry.gatech.edu; aikomari.guzman@upr.edu

The condensation of building blocks into oligomers and polymers is thought to be an early and important process in the origins of life. However, obstacles to the abiotic condensation of amino acids to form peptides include high activation energies, unfavorable thermodynamics, and side reactions. Thioester intermediates in wet-dry cycling provide a possible resolution. Thioesters are condensation products that may have formed en route to peptides during the origins of life[1–17]. A general role for thioesters in prebiotic chemistry was inferred by De Duve, based on their importance and broad distribution in contemporary metabolism[18]. Thioester intermediates enable the anabolism and catabolism of peptides, fatty acids, sterols, and porphyrins[19]. Moreover, thiols are thought to have been abundant on the prebiotic Earth, especially near sources of hydrogen sulfide[20–22].

Peptide bonds were shown to form via thioester intermediates in aqueous media as far back as 1953. Chemically activated building blocks have been used to produce peptides and thiodepsipeptides, which contain both peptide and thioester linkages[23–27]. Wieland and coworkers used valine-thioester and cysteine to form valine-cysteine dipeptides. In those reactions, transthioesterification is followed by an intramolecular thioester-amide exchange between cysteine and a valine-thioester[23]. Weber and Orgel later formed thioesters using an N-acyl protected cysteine or 3-mercaptopropionic acid[24–27]. More recently, attempts to form peptides were made in aqueous solutions via oxidative acylation of thioacids[28–30]. Leman and coworkers showed that carbonyl sulfide can facilitate peptide bond formation via the generation of α-amino acid thiocarbamate, which is cyclized into the highly reactive N-carboxyanhydride[31,32]. Moreover, acetylcysteine or derivative thiols were shown to catalyze peptidyl amidine formation in water when combined with α-aminonitriles[15]. Amide bonds were also shown to form upon reductive acetylation of amino acids with mercaptoacetic acid in aqueous solutions, driven by the formation of pyrite ($FeS_2$) from FeS and $H_2S$[33].

It is known that thiols can condense with carboxylic acids to form thioesters (Fig. 1, top)[18,34]. Further, thioester linkages can be converted to peptide linkages via thioester-amide exchange (Fig. 1, bottom). This reaction scheme is analogous to the acyl substitution scheme that produces depsipeptides, which contain mixtures of peptide and ester bonds. In those reactions, amide bonds form in dry-down reactions of hydroxy acids and amino acids via ester-amide exchange[35–38]. Amide bond formation is enabled by the prior formation of ester bonds. While depsipeptides are a promising prebiotically-relevant approach to peptide bond formation on the ancient Earth, the reactions require mildly acidic pH (optimal around pH ~3.5).

In this work, we report a robust prebiotic system for the formation of peptide bonds in thiodepsipeptides and HS-peptides by reactions of mercaptoacids with amino acids. We hypothesized that kinetic and thermodynamic obstacles to prebiotic peptide bond formation might be resolved by dry-down reactions of mixtures of mercaptoacids and amino acids. In our model, peptide bonds, in oligomers of thiodepsipeptides and HS-peptides (peptides terminated by thiols instead of amines), are produced by reactions with thioester intermediates. Chemically activated building blocks are not required for the scheme; reaction rates and driving forces are modulated by water activity. Our results indicate that thiodepsipeptides and HS-peptides are produced at milder temperatures, under a wider range of pH conditions, and at higher water activity than depsipeptides. These differences can be explained by the higher nucleophilicity of thiols compared to alcohols as well as the higher electrophilicity of thioesters compared to esters[39]. Our results suggest that thiodepsipeptides and HS-peptides could have formed readily on the prebiotic Earth and that thiols acted as key players in the origins of life.

## Results

### Thiodepsipeptides and HS-peptides form via dry-down reactions of mercaptoacids and amino acids.

To test whether thiodepsipeptides and HS-peptides form in dry-down reactions, we dried L-alanine (Ala) with thioglycolic acid (tg) for 1 week at 65 °C under unbuffered, mildly acidic conditions (pH ~3.0) (Fig. 2). Reactions were carried out in an anaerobic chamber to mimic anoxic early Earth conditions and to minimize thiol oxidation. Preliminary studies indicated that oligomer yields were greater when the starting mixture contained an excess of mercaptoacid over amino acid (Supplementary Fig. 1). This difference in yield is likely due in part to an increase in the extent of formation of thioesters with increasing mercaptoacid concentration. Thioesters are required intermediates for thioester-amide exchange. Excess initial mercaptoacid also counteracts effects of its evaporative loss during the drying process, allowing sustained reactions (Supplementary Fig. 2). This model is consistent with a previous investigation of mixtures with varying molar ratios of amino acids and hydroxy acids[37]. In addition, excess mercaptoacids help maintain of a "gel-like" state that enables molecule diffusion throughout the course of the reaction. We therefore employed a 5:1 mercaptoacid/amino acid molar ratio in subsequent reactions. Product mixtures for the reactions were characterized by analytical methods including liquid chromatography-mass spectrometry (LC-MS), direct infusion electrospray high-resolution tandem MS (MS/MS), Fourier transform infrared spectroscopy (FTIR), high-performance liquid chromatography (HPLC), and nuclear magnetic resonance spectroscopy (NMR).

Direct infusion MS and LC-MS analyses indicated that dry-down reactions of tg and Ala produced co-oligomers up to hexamers, with various compositions of tg and Ala (Fig. 2a). Examples of products include 3tg3Ala and 1tg3Ala, with each

**Fig. 1 Proposed acyl substitution reactions for peptide bond formation (amidation) through a thioester-amide exchange.** Under conditions of low water activity, mercaptoacids condense to form thioesters, which can be exchanged for amide bonds in the presence of amino acids. Structures are drawn in their predominant form under acidic dry-down reactions (pH ~3.0).

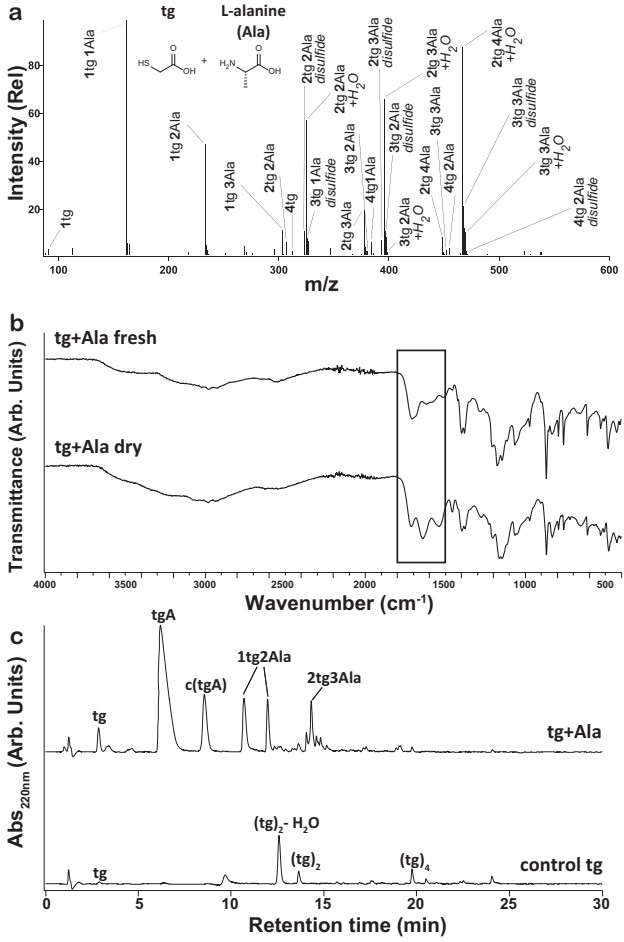

**Fig. 2 Analytical data confirms the formation of thiodepsipeptides in dry-down reactions of tg and Ala. a** Negative-ion mode ESI-MS spectrum of thioglycolic acid (tg) and alanine (Ala) after being dried at 65 °C for 7 days, indicating a variety of thiodepsipeptides. All labeled species correspond to [M-H]⁻ ions. Species that are labeled with $H_2O$ correspond to species that have an additional $H_2O$ unit, which could be the result of non-covalent adducts. **b** Fourier transform infrared spectroscopy (FTIR) shows shifts in the C=O band and in the amide regions upon dry-down of tg and Ala, supportive of thiodepsipeptide formation. **c** HPLC chromatograms of products resulting from drying tg alone or tg with Ala at 65 °C for 7 days. Comparison between the two dry-down reaction products via hydrophobicity-based separation on a C18 column allows the identification of various oligomers that contain Ala. c(tgA) is the cyclic form of tgA, i.e., thiazinedione.

compositional species representing multiple possible sequences. Most oligomer products contained between one to three Ala units, and some observed species contained disulfide bonds (Fig. 2a and Supplementary Fig. 3). We verified that oxidation was minimal under our anaerobic reaction conditions (Supplementary Fig. 4). We confirmed the reproducibility of these results with independent replicate experiments (Supplementary Fig. 5).

**Verification of amide bonds within the product mixtures.** The presence of amide bonds within the product oligomers was confirmed with FTIR (Fig. 2b). The dry-down reaction of tg and Ala resulted in the appearance of amide absorption bands (Amide I and Amide II), direct reporters of amide bond formation[40], as well as changes in the thiol S-H stretch (~2570 cm⁻¹)[41,42] and in the C=O stretch (Fig. 2b)[43–45]. Shifts in the C=O stretch upon dry-down of a control reaction of tg alone, in the absence of

amino acids, support the formation of thioesters by tg (Supplementary Fig. 6)[43–45]. A comparison of the product mixture from dry-down of tg alone with that resulting from dry-down of tg with Ala following separation via hydrophobicity-based C18-HPLC allowed identification of various oligomers that contain Ala (Fig. 2c). We confirmed the identity of a product that is composed of a terminal tg linked with an amide to Ala, herein termed tgA, by comparison to a pure tgA authentic standard (Supplementary Fig. 7). In addition, one of the 1tg2Ala oligomers was also identified by comparison with an authentic standard of a terminal tg linked through two consecutive amide bonds to Ala residues (tgAA, Supplementary Fig. 8). A tgAAA oligomer was also identified via comparison with an authentic standard, although tgAAA is in relatively low abundance (Supplementary Fig. 9). LC-MS also indicated a species with the mass of tgA minus a water molecule, at a retention time of 8.5 min. We hypothesized that this product with a mass corresponding to tgA minus a water molecule is the cyclized form of tgA, i.e., (S)-3-methylthiazine-2,5-dione, herein termed c(tgA). We confirmed this product assignment by synthesizing the (S)-3-methylthiazine-2,5-dione standard and verified that the standard is equivalent to the product species with 8.5 min retention time (Supplementary Fig. 10).

We quantified various products from dry-down reactions using calibration curves of four authentic standards: c(tgA), tgA, tgAA, tgAAA (Supplementary Fig. 11). Our results indicate that 45% of Ala is converted into tgA, 3% into c(tgA), and 6% into tgAA. Products containing sequential amide bonds, such as the tgAA molecule, can form through thioester intermediates, for instance between tg and tgA, as illustrated in Supplementary Fig. 12. The existence of thioester-containing compounds was verified via collection of full UV-VIS spectra during HPLC analysis, which demonstrated the existence of various peaks with the typical thioester absorbance peak at ~235 nm (Supplementary Fig. 13)[46,47]. Incubation in the water of a previously dried mixture of tg and Ala for 3 days at 65 °C under anoxia led to partial degradation of various oligomers, further supporting the existence of thioester bonds among the dry-down products (Supplementary Fig. 14). Time-course analysis of dry-down reactions shows gradual oligomer formation by tg and Ala for up to 7 days with an initial rise in the thioester-containing tg dimer that is gradually consumed by conversion to other products (Supplementary Fig. 15).

For a more detailed mechanistic investigation using plausible reaction intermediates, we performed reactions in which we used either a tgA standard (containing an amide bond), an Atg standard (containing a thioester bond), or thiazinedione c(tgA) in the presence or absence of either tg or Ala. We suspected that c(tgA) can undergo ring-opening polymerization with tg or Ala (Supplementary Fig. 16)[48–50]. To test this hypothesis, we dried c(tgA) in the absence or presence of tg or Ala. Indeed, we observed the fast formation of products following dry-down that would be expected to result from ring-opening polymerization, such as tgAA (Supplementary Fig. 17). A notable thioester hydrolysis product was observed as well, as some tgA also occurred following drying of c(tgA) at 65 °C (Supplementary Fig. 17). On the other hand, reactions involving the Atg standard showed fast thioester hydrolysis followed by the formation of tgA from free tg and Ala (Supplementary Fig. 18), whereas reactions of tgA showed the formation of species that indicated some amide hydrolysis of the tgA standard (e.g., the tgAA product was observed, Supplementary Fig. 19).

To confirm amide bond formation during dry-down reactions of amino acids with mercaptoacids, dried mixtures were resolved by C18-HPLC, and isolated peaks were collected for analysis via direct infusion electrospray high-resolution tandem MS (MS/MS)

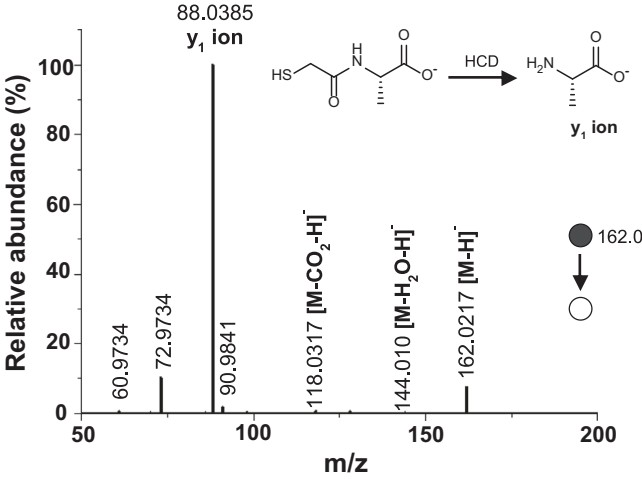

**Fig. 3 Amide bonds form upon dry-down of amino acid and a mercaptoacid as indicated by MS/MS Higher-energy C-trap dissociation (HCD) analysis.** tg and Ala were dried at 65 °C for 7 days; the resulting products were analyzed by negative-ion mode MS/MS. Tandem MS analysis was conducted using $m/z$ 162.0 as the precursor ion. Negative-ion mode MS/MS analysis of the [tg+Ala-$H_2O$] molecule showed a base peak at $m/z$ 88.0385, being the $y_1$ ion of the sequence tgA. This fragmentation pattern is supportive of the formation of amide bonds in these dry-down reactions.

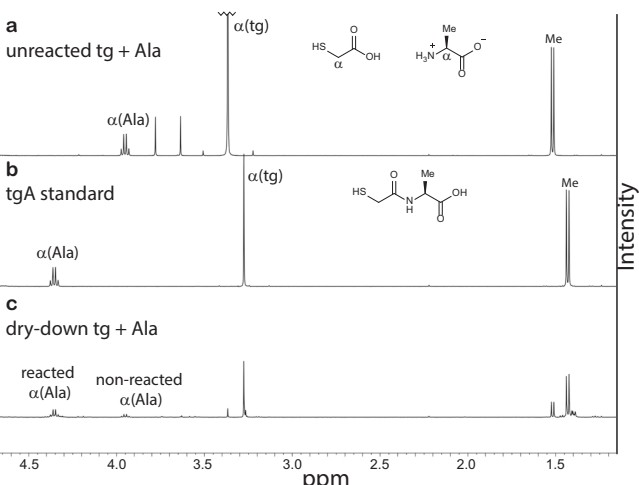

**Fig. 4 Dry-down reactions of tg and Ala produce oligomers containing amide bonds.** $^1$H NMR spectra of: **a** A fresh, unreacted mixture of thioglycolic acid (tg) with alanine (Ala). **b** The authentic standard of Ala acylated with tg at the α-amine (tgA) demonstrating the down-field shift of the α-resonance upon amidation of Ala. **c** Sample of tg dried with Ala in a 5:1 molar ratio at 65 °C for 7 days. The starting tg material contains some thioester-linked dimer (tg)$_2$. c(tgA) is the cyclic form of tgA, i.e., thiazinedione.

of tg and Ala dry-down reaction products. As mentioned above, identification of the tgA product peak was accomplished using an authentic standard, and the presence of an amide bond in this product was verified with MS/MS (Fig. 3). The tgA sequence was validated by the small mass error, −1.3 mDa, the MS/MS mercaptopeptide fingerprint, water and $CO_2$ neutral losses, and $y_1$ ion detection for the Ala residue in the C-termini. Further structure validation was performed by comparing the tgA MS/MS spectra with that of the tgA authentic standard, which showed almost identical fragment ions (Supplementary Fig. 20a). Characterization of other peaks by MS/MS identified the formation of additional amide bonds in other sequences (Supplementary Fig. 20b). For example, fragment ions observed for $m/z$ 233.1 MS/MS analysis correspond to the sequence tgAA (Supplementary Fig. 20b).

**NMR characterization of product mixtures using authentic standards.** NMR characterization of dry-down reaction products supported the high conversion of the building blocks into oligomers. The reaction of tg with Ala resulted in a relatively simple set of chemical envelopes in $^1$H NMR spectra that were interpreted by comparison to spectra of the pre-dried mixture and the tgA authentic standard (Fig. 4 and Supplementary Figs. 21–27). In unreacted Ala, the resonance of the α-proton has a chemical shift of 3.95 ppm (Fig. 4a). After dry-down of Ala with tg, the intensity of the 3.95 ppm resonance decreases, and three envelopes of product α-protons were observed, centered at 4.18, 4.35, and 4.52 ppm. The identities of these resonances as α-protons of reacted Ala species were supported by $^1$H-$^1$H COSY NMR (Supplementary Fig. 23). 77% of the α-protons of Ala shifted down-field to 4.35 ppm, in accordance with the chemical shift of the α-proton in the tgA standard (Fig. 4b, c). Notably, in the reaction products there are many overlapping resonances in the 4.35 ppm envelope (Supplementary Fig. 25), which arise from amidated Ala species. Among the other product α-protons in this envelope are those that belong to tgAA, as confirmed by the tgAA standard (Supplementary Figs. 26, 27). Approximately 3% of the

α-protons of Ala shifted down-field to 4.18 ppm, and an additional 3% are located at the 4.52 ppm region after the dry-down reaction. Integration of the free, unreacted α-proton resonance of Ala indicated that 83% of Ala was converted into products after 1 week at 65 °C.

**Formation of proto-peptides under a wide range of pH and temperature conditions.** To investigate the generality of thiodepsipeptide and HS-peptide formation via drying mercaptoacids with amino acids, we combined either tg or thiolactic acid (ta) with various amino acids, including Ala, glycine (Gly), phenylalanine (Phe), or cysteine (Cys). Analysis of these dry-down reactions via both MS and HPLC confirmed the formation of product mixtures that were consistent with oligomer formation (Supplementary Figs. 28–37). As expected, dry-down of the amino acids in the absence of mercaptoacids did not produce oligomers (Supplementary Figs. 29, 31, 33, 35).

The reactions described above were carried out using unbuffered, mildly acidic conditions (pH ~3). Based on the nucleophilicity of thiol moieties we postulated that thiodepsipeptides and HS-peptides could form at pHs closer to neutrality. To test this possibility, we carried out dry-down reactions containing mixtures of tg and Ala in which the initial reaction pH was adjusted to 3.5, 5.5, 6.5, or 7.0. One, 5, 10, or 20 equivalents of imidazole, with respect to the amount of amino acid, were added to adjust the pH. The addition of imidazole to the reactions helped maintain a "gel-like" state, which is likely to enable diffusion of the molecules during the reaction. HPLC and MS analyses indicated that thiodepsipeptide formation was most efficient under acidic conditions. Nevertheless, thiodepsipeptides were observed in good yields at every pH tested (Fig. 5a and Supplementary Fig. 38). Specifically, $^1$H NMR analysis indicated that 90% of Ala was converted into products upon dry-down with tg at pH 3.5, 89% was converted at pH 5.5, 71% at pH 6.5, and 42% at pH 7.0 (Supplementary Figs. 39–42). To test for thiodepsipeptide formation at basic pH, we carried out dry-down reactions containing mixtures of tg and Ala in which the pH was adjusted using NaOH (Supplementary Fig. 43). Although

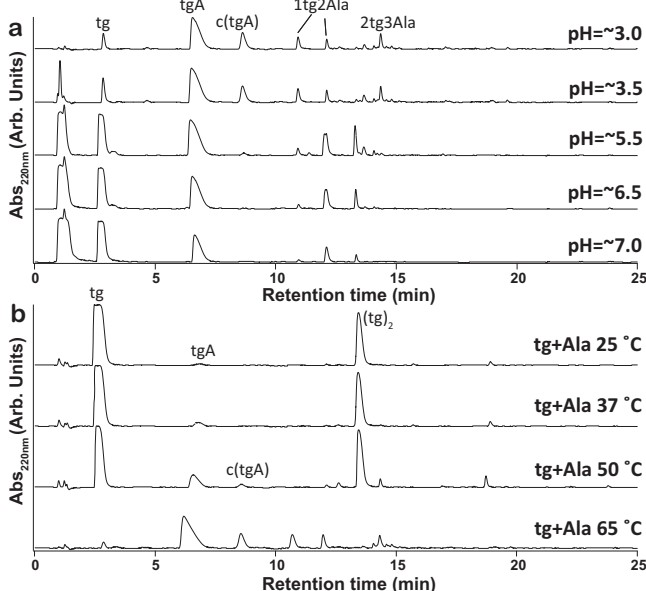

**Fig. 5 Thiodepsipeptides form under a range of pH conditions and temperatures. a** Chromatograms of products resulting from thioglycolic acid (tg) and alanine (Ala), in a 5:1 molar ratio, being dried at 65 °C for 1 week in various initial reactions pH. Products are observed throughout the whole range of pH conditions tested. **b** Chromatograms illustrating the temperature dependence of thiodepsipeptide formation under unbuffered, acidic conditions (pH ~3). Products are detected when mixtures of tg and Ala are dried and maintained at temperatures as low as 25 °C. (tg)$_2$ refers to a thioester-containing homodimer of tg. c(tgA) is the cyclic form of tgA, i.e., thiazinedione.

yields were low, some product formation was still observed at pH 9.0 and even at pH 12.5 (Supplementary Fig. 43).

While we initially focused on the formation of thiodepsipeptides and HS-peptides under evaporative dry-down conditions, which shift the thermodynamic equilibrium towards condensation-dehydration products, we postulated that thiodepsipeptide formation might also be possible to some extent in systems with high water activity. Indeed, following incubation of a mixture of tg and Ala at 65 °C for 1 week in water in a closed reaction vial, various product peaks are evident, some of which contain amide bonds, although the extent of reaction is lower in high water activity than in dry-down reaction conditions (Supplementary Fig. 44).

Thiodepsipeptide formation is temperature-dependent and occurs under relatively mild temperatures, implying relatively low activation energies. Products were detected in dry-down reactions containing mixtures of tg and Ala at temperatures as low as 25 °C (Fig. 5b). NMR analysis indicated that drying Ala with tg and maintaining the sample in the dry state for 1 week resulted in the conversion of 4% Ala into oligomers at 25 °C, while 6% was converted at 37 °C, 23% at 50 °C, and 83% at 65 °C (Fig. 5b and Supplementary Figs. 45–47).

**Comparison of thiodepsipeptide and depsipeptide formation**. We have observed the formation of proto-peptides, driven by low water activity, in mixtures of mercaptoacids and amino acids under a wide range of pH and temperature conditions. We directly compared the formation of thiodepsipeptides and HS-peptides through dry-down reactions of mercaptoacids and amino acids with analogous reactions for depsipeptides formation from mixtures of hydroxy acids and amino acids[35–38]. Both depsipeptide and thiodepsipeptide polymerization occur under acidic conditions. In contrast to thiodepsipeptides, no depsipeptide products formed at

pH 7 (Supplementary Fig. 48). The activation energies for the formation of thiodepsipeptides are lower than for depsipeptides. While products were detected in dry-down reactions containing mixtures of tg and Ala even at temperatures as low as 25 °C (Fig. 5b), no products were observed under these conditions for dry-down reactions containing the equivalent hydroxy acid and amino acid mixture (Supplementary Fig. 49). In contrast to thiodepsipeptide formation, which proceeded to some extent in high water activity (Supplementary Fig. 44), incubation in high water activity of a mixture of glycolic acid (glc, the hydroxy acid analog of tg) with Ala at 65 °C for 1 week did not produce oligomers (Supplementary Fig. 50). The extent of conversion of a given amino acid into products is higher when dried-down with mercaptoacids compared to when dried-down with hydroxy acids. Specifically, 83% of Ala was converted into products when dried-down with tg, whereas only 64% of Ala was converted into products with glc (Fig. 4 and Supplementary Fig. 51).

## Discussion
We describe the formation of thiodepsipeptides and HS-peptides by dry-down reactions of mercaptoacids with amino acids. Product mixtures from these reactions were characterized by a variety of methods, including liquid chromatography-mass spectrometry (LC-MS), direct infusion electrospray high-resolution tandem MS (MS/MS), Fourier transform infrared spectroscopy (FTIR), high-performance liquid chromatography (HPLC), and nuclear magnetic resonance (NMR). We compared the reaction products of mercaptoacids with amino acids to the reaction products of hydroxy acids with amino acids. Reactions of mercaptoacids with amino acids allowed the formation of thiodepsipeptides and HS-peptides under wide ranges of pH (up to pH 7.0 and temperature down to 25 °C). Reactions of hydroxy acids with amino acids to form depsipeptides require more acidic conditions and higher temperatures.

The reactions reported here were conducted under anoxia to minimize the oxidation of thiol-containing compounds. The anaerobic conditions in our experiments are plausibly prebiotic, mimicking the atmosphere of the prebiotic Earth, long before the great oxidation event around 2.4 Ga[51]. As a control, we dried-down thioglycolic acid with alanine under atmospheric oxic conditions and observed the formation of a variety of side products, including those containing disulfide bonds (Supplementary Fig. 52). This result confirmed the importance of anaerobic conditions in experiments designed to explore the possibility of thioester-mediated peptide bond formation in an early Earth environment.

To examine the monomer exchange of thiodepsipeptides and HS-peptides under a plausible geochemical scenario of cycling environmental conditions, we performed iterative dry–wet reactions in addition to one-step dry-down reactions. We anticipate a driving force for condensation-dehydration during the dry phases and a driving force for hydrolysis during the wet phases. The results show that thiodepsipeptides are formed by cycling, albeit with a lower yield than in one-step dry-down reactions (Supplementary Fig. 53). It is possible that oligomers produced in our reactions were reduced in length and yield by a "backbiting" mechanism, in which a terminal thiol performs an intramolecular nucleophilic attack on a thioester bond to produce a six-membered ring c(tgA).

In conclusion, we have presented a plausible prebiotic scenario in which thiodepsipeptides and HS-peptides are formed over a wide range of conditions: varying hydration levels, temperature, and pH. Previously, short thiol-terminated compounds and thioesters were investigated in both aqueous and non-aqueous solutions as building block units for dynamic combinatorial

libraries[48,52–55]. Based on the results presented here we propose that the incorporation of even simple mercaptoacids into these systems might offer new avenues for dynamic combinatorial chemistry in water or under dry-down conditions. Of importance to origins of life research, complex mixtures of building blocks, including amino acids, mercaptoacids, and hydroxy acids, are arguably more prebiotically plausible than a single purified class of molecules, such as amino acids. Taken together, the results of our study suggest that HS-peptides and thiodepsipeptides could have formed on prebiotic Earth and participated in chemical evolution, ultimately giving rise to life.

## Methods

**Authentic standards**. The synthesis of authentic standards is presented in the Supplementary Information.

**Dry-down reactions**. All dry-down reactions that contained mercaptoacids were carried out in a Coy anoxic chamber (97% Ar and 3% $H_2$ headspace) to minimize oxidation, unless stated otherwise.

For polymerization experiments, aqueous solutions of mercaptoacids and amino acids (typically 166.66 mM amino acid and 833.33 mM mercaptoacid) were allowed to dry in Eppendorf tubes at various temperatures in a dry block heater (25, 37, 50, or 65 °C), with caps open, for various durations. Prior to analysis, samples were resuspended with 90% ultrapure water, 10% acetonitrile to a given concentration, vortexed, and sonicated.

For polymerization experiments containing imidazole, an imidazole solution was aliquoted and lyophilized prior to the addition of thioglycolic acid and alanine in the anaerobic chamber.

For polymerization experiments of control depsipeptides, aqueous solutions of glycolic acid and amino acids were allowed to dry in Eppendorf tubes at various temperatures in an oven or on a dry block (25, 37, 50, 65, or 85 °C).

For dry-down experiments of thiodepsipeptides and HS-peptides under oxic conditions, thioglycolic acid and L-Ala were allowed to dry in an oven at 65 °C for 7 days.

For the variable pH dry-down experiment, samples were dried under different pH conditions by the addition of a different number of equivalents of imidazole or NaOH. For Imidazole-containing reactions, the initial reaction pH was adjusted to 3.5, 5.5, 6.5, or 7.0. One, 5, 10, or 20 equivalents of imidazole were added with respect to the amount of amino acid to adjust the pH. For NaOH-containing reactions, the initial reaction pH was adjusted to 3.5, 9.0, or 12.5. Two, 5, or 10 equivalents of NaOH were added with respect to the amount of amino acid to adjust the pH. pH was measured before and after the reactions. The pH did not fluctuate more than 0.5 pH units after the reaction. For the pH-dependent experiment in solution, samples were incubated in water (pH ~3.0, 55.55 mM Ala and 277.75 mM tg) or in the presence of 20 eq $Na_2HPO_4$ relative to the amount of amino acid (pH ~7.0, 38.46 mM Ala and 192.30 mM tg).

For the dry–wet cycling experiment, aqueous solutions of mercaptoacids and amino acids were allowed to dry in Eppendorf tubes at 65 °C in a dry block, with the caps open. After 24 h samples were cooled back to room temperature, resuspended with 200 μL of $H_2O$, mixed, and were allowed to dry with an open cap. This procedure was repeated five more times for a total reaction time of 1 week.

For dry-down reactions involving varying amounts of tg and Ala at a 1:1 molar ratio, tg and L-Ala were dried at varying amounts (from 10 to 250 μmol) at 65 °C for 7 days. Samples were rehydrated with appropriate volumes to give 100 mM Ala concentration (starting Ala concentration).

For dry-down reactions involving the tgA, Atg, or c(tgA) standards, the standards were dry-heated at 65 °C for up to 7 days in the absence or presence of either alanine or tg at a 1:10 molar ratio (in favor of either tg or alanine).

**Direct infusion mass spectrometry**. All peak assignments correspond to [M-H]⁻ ions unless otherwise noted. Samples were directly infused into a mass spectrometer using the parameters: solvent: 95% $H_2O$, 5% acetonitrile. Flow rate: 0.5 mL/min. Five microliters injection with $H_2O$ needle wash. UV detection at 210, 257, or 280 nm. 0.6 cm path length. Scanning ±65 to ±2000 $m/z$. Equipment: ESI-MS— Agilent 6130 single quadrupole MS (Agilent Technologies, Santa Clara, CA) with UV detector coupled to Agilent 1260 HPLC. Capillary voltage: 2.0 kV. Fragmentor voltage: 70 V. Processing of MS data were conducted using a suite of macros within Igor Pro 8.0 (Wavemetrics)[37].

**High-resolution MS and MS/MS analysis**. For direct infusion high-resolution analysis, the fractions collected for different important chromatographic regions were diluted 1 to 10 in a 1:1 acetonitrile-water solution. Tandem MS analyses were performed using a Q-Exactive hybrid quadrupole-Orbitrap mass spectrometer (Thermo Scientific) operating with a heated electrospray interface (HESI). Analyses of products from dry-down reactions were carried out in negative-ion mode with the following operating conditions: maximum ion injection time (50 ms), FT

resolution (17,500), AGC target (1e6), HCD energy (NCE = 35), $MS^2$ isolation width $m/z = \pm1.5$, voltage −2.8 kV, capillary temperature (320 °C), sheath gas (10 L/min), auxiliary gas (2 L/min), S-lens RF level 50. The Q-Exactive mass spectrometer was calibrated using Pierce™ Negative Ion Calibration Solution (Thermo Scientific). Data were processed using Xcalibur™ version 4.0.

**High-performance liquid chromatography**. HPLC analyses were conducted using ChemStation on an Agilent 1260 quaternary pump and Agilent 1260 Autosampler with DAD UV-vis detector, with a path length of 1.0 cm. Samples were separated using a Phenomenex Kinetex 2.6 mm xB-C18 100 Å, LC column 150 × 2.1 mm. Column temp: 25 °C. 10 μL Injection. Solvents: Solvent gradient was: (A) 0.1% formic acid in LC-MS grade water, (B) LC-MS grade acetonitrile. Flow Rate: 0.3 mL/min. Gradient: 5 min 100% A, 0% B; 20 min ramp to 45% A, 55% B; 10 min 0% A, 100% B; 1 min ramp 100% A, 0% B; 14 min 100% A, 0% B. Wavelengths monitored: 210 and 220 nm, with entire spectrum 180–400 nm collected in 2 nm steps. Processing of HPLC data were conducted using either Excel or a suite of macros within Igor Pro 8.0 (Wavemetrics).

**Liquid chromatography-mass spectrometry**. HPLC equipment and chromatography conditions: Agilent 1290 HPLC pump and thermostat. Agilent 1260 Autosampler and DAD UV-vis detector with a path length of 0.6 cm. Agilent 1260 quaternary pump and RID. Column: Phenomenex Kinetex 2.6 mm xB-C₁₈ 100 Å, LC column 150 × 2.1 mm. H15-191145, 5603-145. Column temp: 25 °C. Ten μL injection with needle wash, 100 μL/s injection speed. Solvents: (A) 0.1% formic acid in LC-MS grade water, (B) LC-MS grade acetonitrile. Flow rate: 0.3 mL/min. Gradient: 5 min 100% A, 0% B; 20 min ramp to 45% A, 55% B; 10 min 0% A, 100% B; 1 min ramp 100% A, 0% B; 9 min 100% A, 0% B. Wavelengths recorded: the entire spectrum from 180 to 400 nm was collected in 2 nm steps with realtime monitoring of absorbance at 210 and 220 nm. This system was coupled to an Agilent 6130 single quadrupole mass spectrometer using the following parameters: Scanning ±65 to ±2000 $m/z$. Capillary voltage: 2.0 kV. Fragmentor voltage: 70 V. Single ion monitoring was carried out using the same system and tracking particular ions (as listed in Fig. S32).

**NMR spectroscopy**. Samples were dissolved in a phosphate buffer (pH 2.5) in $D_2O$ and $^1H$ NMR spectra were recorded using Bruker Avance III-HD-500 MHz spectrometer and TopSpin. The temperature was 298 K, using a t1 relaxation delay of 15 s, collecting 64 scans. All spectra were processed and plotted using the MestReNova Software. The overall conversion of amino acid monomer into polymers was estimated from the integration of the free, nonamidated α-proton $^1H$ NMR resonance.

**IR spectroscopy**. IR data was obtained on a Thermo Nicolet 4700 FTIR spectrometer. Prior to analysis, samples (10 μl, 50 mM amino acid, or 250 mM mercaptoacid monomer) were placed on Durapore® hydrophobic PVDF membranes with a pore size of 0.22 μm (#GVHP04700, Millipore Sigma) and allowed to dry. Dried samples were analyzed in an Attenuated Total Reflectance (ATR) sample chamber. Spectra were background-subtracted from 400 to 4000 cm⁻¹ and signal-averaged (16 scans per spectrum).

**Reduction with tris(2-carboxyethyl)phosphine**. To analyze for oxidation levels, we incubated a dried mixture of tg and Ala with 500 mM tris(2-carboxyethyl) phosphine (TCEP) for 1 h at RT. The resulting products were analyzed by hydrophobicity-based separation using C18-HPLC.

**Hydrolysis assay**. We incubated a previously dried mixture of tg and Ala in water for 3 days at 65 °C (unbuffered, pH ~3.5) under anoxic conditions. Samples were analyzed by hydrophobicity-based separation using C18-HPLC before and after incubation in water.

**Reporting summary**. Further information on research design is available in the Nature Research Reporting Summary linked to this article.

## Data availability

All the data supporting the findings of this study are available within the main text and its Supplementary Information.

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

## Acknowledgements

We thank Drs. Arthur L. Weber, Ramanarayanan Krishnamurthy, Jay Forsythe, Anton Petrov, and David Fialho for fruitful discussions. We acknowledge Dr. Milan Gembicky in the UCSD crystallography facility for the determination of the crystal structure of (S)-3-methylthiazine-2,5-dione. This research was supported by the NSF and the NASA Astrobiology Program under the NSF Center for Chemical Evolution [CHE−1504217, N.V.H.].

## Author contributions

M.F.-P., N.V.H., and A.G.-M. conceived and designed the experiments. M.F.-P., A.G.M., and M.B. carried out the experiments. M.F.-P., M.B., F.M.F., L.J.L, L.D.W., N.V.H., and A.G.-M. wrote the paper. L.J.L. synthesized authentic standards. M.F.-P., M.B., F.M.F., L.J.L., L.D.W., N.V.H., and A.G.-M. contributed to the data interpretation. N.V.H. and A.G.-M. supervised the research. All authors reviewed the paper.

## Competing interests

The authors declare no competing interests.

**Additional information**

