## [Peer Review File · Nature Communications]

REVIEWER COMMENTS

Reviewer #1 (Remarks to the Author):

Freckel-Pinter et al. report that when thiols such as thioglycolic acid, tga, and an amino acid are heated thiopeptides are formed. They suggest 'thiol catalyzed' peptide formation is observed. Glycine, alanine, phenyl alanine or cysteine are used as the amino acids. Excess thiol promotes the reported reactions, and 5:1 tga to amino acid is suggested to be optimal. Little conversion is observed with 1:1 thiol and amino acid. Excess amino acid is not investigated. The authors do not explain why the amino acid inhibits these reactions, but this likely indicates products are largely tga-oligomers or oxidation products, as shown in Supplementary Information, but the product distribution is unclear.

The analysis is qualitative, and products are assumed based on relative binding to C18 silica. The authors only address "combined yields" for total observed products in comparison to the residual amino acid. The authors also do not provide data on the hydrolysis of the reaction products, which would remove all suspected thioesters shown in Supplementary Fig. 2. The authors also do not reduce the disulphide products after the reaction, which would remove all suspected disulphides, shown in Supplementary Fig. 2. If oxidation occurs, how do the authors account for this?

Miller has already describe the effect reported here. In 1995 (Nature 373, 683-685 (1995)), Miller reported dry heating amino acids with thiols gives amides and peptides, "at 40 °C the lowest temperature used so far in which amide bonds are formed". Miller's work was also presented in the context of the origins of life. Miller's paper is cited in a reference string in the introduction, but its outcomes are not explained.

The title of this paper should specify thiopeptides not peptides.

Freckel-Pinter et al. state: "We report robust, plausible prebiotic reactions of mercaptoacids with amino acids that result in the formation of peptides and thiopeptides, which contain both peptide and thioester bonds." What is meant by "robust", how is this quantified? Are any pure peptide (only alanine products) formed or only thiopeptide (tga + alanine)? Do all products contain tga (thiol or disulfide)? What fraction of new bonds are peptide (alanine-alanine) and what fraction are thiopeptide (tga-alanine or alanine-tga)? What fraction of amides are AA-amides and what fraction are tga-amides? What fraction of new bonds are thioester vs amide bonds? This could be quantified.

The authors should demonstrate what they have synthesised and quantify specific products – in a heterogeneous reaction this is important. Demonstrating the product distribution is especially important when the core discussion is focused on coupling amino acids, and suggests tga is a “catalyst”.

Is tga catalytic? What turn over number is observed for tga in these reactions?

As the work is presented the “tga-amide bonds” (in for example, tga-A) are used to suggest that high “peptide yields” are observed, but this compound is not a peptide. Most of the suggested products (Supplementary Fig 2) don’t contain peptide bonds – this is not clear in the manuscript.

Proposed peptide products, such as the alanine tetramer, are commercial and could easily have been verified by the authors.

Freckel-Pinter et al. suggest a wide range of pHs work. But the range is restricted and does not include any pH above neutral. It is also not clear that the dry reactions are at neutral pH after drying. How is the pH measured or calculated in the dry reactions? What happens at higher pH? How do the authors verify that under the dry conditions the “pH” of the reaction is not acidic. Removing water (volatiles) during drying will significantly shift the pH. For example, Miller (see Nature 373, 683-685 (1995)) reports that solutions of amino acids and thiols at initial pH 7, following drying dropped to pH 4.

The suggested important distinction between this work and the previous work (with esters rather than thioesters) is that the new work is no longer restricted to “acidic conditions”. This needs to be more rigorously proven. The authors also need to specify the pH range that the previous ester mediated synthesis was restricted to.

In the “dry” samples, with 5 eq. tga, what is the bulk material? Powder or liquid? Is the reaction being run in liquid tga?

Freckel-Pinter et al. suggest their reaction is the “most robust one-pot pathway for peptide formation ever reported”. This comment is totally unjustified. How have the authors quantified the robustness of their reaction against other reported peptide syntheses. What even makes this process “one-pot”? What does that mean? They have to separately synthesise and isolate tga and

alanine, and then mix them, why is this “one-pot”. Again, what is robust here – the authors keep repeating this term, but it is meaningless. The reaction is certainly not robust to something as simple as the “wrong” amount of amino acid. It is shut down by more amino acid. This is not a robust reaction.

Freckel-Pinter et al. should cite Foden et al. *Science*, 2020, 370, 6518, 865-869, that demonstrates tga, and other “simple thiols such as coenzyme M (79%) (fig. 85)”, catalyze peptide formation in water at neutral pH from nitrile and amine reactions. This work was also presented in the context of the origins of life.

Freckel-Pinter et al. suggest they observed high conversion to “oligomers”, this is exaggerated by including the single reaction of tga and amino acid in the yield of “oligomers”. This is not an oligomer, it is not even a dimer, it is the product of a reaction between two different compound types. The authors could easily deconvolute the yield of this single major reaction product from the “oligomer” yield. The “oligomers” also appear to incorporate significant numbers of disulphides, these compounds are likely exaggerating the “oligomer” yield which is presented as amide or peptide or thiopeptide in the paper.

What happens if “longer” oligomer products (or suspected products) are subjected to the conditions of the reaction? Are they stable? Most of the products shown in Supplementary Fig. 2 are expected to be significantly less stable than peptides.

Fig. 5. What is “tgA” vs “tgA-H₂O”? What is the structure of tgA-H₂O?

Fig. 5. What are the major peaks at 3 mins and 15 mins at 25 – 50 °C? Why do both disappear at 65 °C? If this is loss of a volatile – how are yields accurately quantified?

Freckel-Pinter et al. state: “Kinetic analysis shows gradual oligomer formation by tga and Ala for up to seven days in the dry state (Supplementary Fig. 7).” Why is this a kinetic analysis? It is just a time course. Rates could not be extracted from the data provided. The authors also state, with no evidence, that gradual oligomer formation occurs? How do they prove this? How do they quantify the total content of material – this is not clear.

Over seven days the author show that tgaA grows significantly. But it is not clear that the other product are not due to, for example, oxidation? Why can’t the data acquired in Supplementary Figure 10 and Figure 13, and HPLC of the corresponding samples (e.g. Fig. 5) be used to quantify the

amount of tgaA? The same could be done for tgaAA (see Figure 6). This data could then be used to assess the absolute conversion and relative yields.

Freckel-Pinter et al. report LCMS. The authors should be able to calculate how much peptide and how much thiodepsipeptide of each composition is present in the mixture. Why don't they?

Freckel-Pinter et al. interpret NMR peaks as "oligomers", which they then infer to be "peptides", when they have not quantify the content of tgaA. The simplicity suggests the product is largely one compound. The NMRs all lack the complexity expected for significant amounts of oligomer products.

Why do the authors show 1H-1H NMR, that simply verifies the Ala connectivity of the starting material. Why don't the authors provide 1H-13C NMR that could be used to verify the reactions discussed.

What are the 1H-1H signals between peaks at 3.5 ppm - 4.2 ppm? These are not identified. They are not alanine peaks, what are they? They are relatively major products, likely second major product, after tgaA.

Supplementary Fig 19 is mislabelled, both traces are labelled as the control.

Supplementary Fig 20: what is the large signal at 2.1 ppm after dry down? It is a major component but does not apparently relate to tga or cysteine. What are the large signals in the starting materials at 3.8 ppm and 3.6 ppm? Cysteine is a thiol-acid. Why is tga needed in this reactions?

Supplementary Fig 25: why is the spectrum cut at 3.0 ppm? Is the unexplained peak from Fig 20 here too? As above, what are the large signals in the starting materials at 3.8 ppm and 3.6 ppm? The authors state 66% of Phe has converted into oligomer, but what fraction of this is Phe-Phe peptide and what fraction is just tga-Phe?

Why is imidazole used in the "pH" range? What happens if the pH is adjusted with NaOH rather than imidazole? How do the authors ensure the pH is not changed during dry down. Are the pH values measured or calculated? What do the authors mean by "referring to the number of eq. of acid"? Do they mean 20 eq. for tga and 20 eq. for Ala? If so this is 40 eq. Are the pH values of these reactions checked before and after drying? If so they should be reported, if not, why not?

Supplementary Figure 31: What concentration are the compounds? The authors state “Various product peaks are evident even in the wet phase, some of which contain amide bonds (such as tgaA).” What is meant by “wet phase”? What is the concentration? Is this a solution? The authors don’t provide a “pH” for the reaction in “wet phase”, they need to give this pH. Have the authors demonstrated that this “wet” reaction yields peptides (i.e. AA bonds)? This should be clarified in the text. The only labelled peak doesn’t contain peptide, only tgaA. What are the other “peaks”? What are the peaks at 3 mins, 12 mins, and 14 mins?

Low “yields” from wet-dry cycles may indicate the instability of the products, casting further doubts on the claim of robustness.

Reviewer #2 (Remarks to the Author):

- What are the noteworthy results?
- Will the work be of significance to the field and related fields? How does it compare to the established literature? If the work is not original, please provide relevant references.
- Does the work support the conclusions and claims, or is additional evidence needed?
- Are there any flaws in the data analysis, interpretation and conclusions? - Do these prohibit publication or require revision?
- Is the methodology sound? Does the work meet the expected standards in your field?
- Is there enough detail provided in the methods for the work to be reproduced?

Frenkel-Pinter et al. report on the reaction of amino acids with thiols. The authors show that by drying these compounds together, two key functional groups of metabolism: thioesters and peptide bonds can be formed. Furthermore, the authors show that the thiopeptides also form in water, which is surprising and extends the relevance of the work. Finally, the authors are able to compare their results with previous results from their groups, highlighting the key difference of sulfur in a thiol as a nucleophile in comparison to a hydroxyl group with the formation of peptides: peptides are unable to form at neutral pH, at low temp., and not in water, which is in contrast to the thiol derived results presented. Altogether these data are a significant advance.

The writing is clear and the results are fairly straightforward. The results are original and will be important not only for work in the origins of life field but also as written by the authors, more broadly for example in the study of combinatorial libraries, and even possibly materials chemistry. There are no major flaws in the interpretations or claims made by the authors that I see, and the work is reported in sufficient detail.

I have a few small comments that I hope might be of use for the authors to improve the article.

Figure 1) I'd like to suggest the authors add some more material here. Specifically, how about adding a reaction showing the formation of the aminoacid-thioester, which is presumably needed to form dipeptides such as the tg-AA? In the current form, I may lack the needed chemical imagination but I stumbled on how to go from the tg-amide to an amino acid dimer. I think adding at least this one more reaction would be useful, and the authors might consider adding others, which would elevate the work and make it more interpretable.

Figure 2) (and elsewhere) in the figure structures of fully protonated tg is drawn, and also carboxy-protonated alanine. Are they the correct forms under negative ionization mode? please confirm.

Figure 2C) Whereas thioesters show an absorbance feature at ~230nm, peptides show an absorbance at 280nm. It could be useful to show two traces here, one at each wavelength.... just an idea; up to the authors.

SI figures 5 and 6. It might be best to stack the traces on top of each other, since they align so closely that it's hard to see each one.

Supplemental figure 7. Can the authors identify the peaks? Additionally, in these traces (as well as elsewhere) some peaks increase and some decrease during the course. Can the authors comment on that?

Response to reviewers

Reviewer 1:

Reviewer 1, Comment 1. *Freckel-Pinter et al. report that when thiols such as thioglycolic acid, tga, and an amino acid are heated thiopeptides are formed. They suggest ‘thiol catalyzed’ peptide formation is observed. Glycine, alanine, phenyl alanine or cysteine are used as the amino acids. Excess thiol promotes the reported reactions, and 5:1 tga to amino acid is suggested to be optimal. Little conversion is observed with 1:1 thiol and amino acid. Excess amino acid is not investigated. The authors do not explain why the amino acid inhibits these reactions, but this likely indicates products are largely tga-oligomers or oxidation products, as shown in Supplementary Information, but the product distribution is unclear.*

Response: We thank the reviewer for the thoughtful comments and suggestions, which have helped us improve our paper and clarify important aspects of our study.

In the revised manuscript we have included data from new experiments that provide additional support for the proposed thioester-amide mechanism and that confirm the presence of amide bonds in the products of these reactions. In the first series of these experiments, the products of constant amino acid concentration (Ala) and varying concentration thiol (tg) reactions demonstrate that increasing tg increases the rate of tgA formation (which contains an amide linkage) (Now included as Supplementary Figure 1). This data shows that increasing the mercapto acid concentration increases the rate of amide bond formation, supporting the role of thiols in catalyzing amide bond formation.

Reviewer 1’s impression that amino acids inhibit mercapto acid-mediated peptide bond formation is understandable given the data presented in our original manuscript. This impression appears to be due to a subtlety of our peptide formation reaction. Thioglycolic acid (tg) has considerable volatility at the temperatures used for proto-peptide synthesis (tg boiling point is 96 °C). This volatility results in a decrease in the tg concentration over the course of dry-down reactions, which explains the almost complete lack of reactivity for the 0.2:1 reactions but the lack of significant differences in product yield between 1:1 and 0.5:1 reactions (new Supplementary Figure 1). To further emphasize the impact of tg volatility on product yields, in a second series of new experiments we kept the amino acid to thioglycolic acid ratio the same (1:1), but varied the total amount of acids, ranging from 10 µmol to 250 µmol. As evident in the new Supplementary Figure 2, there is a positive correlation between the amount of mercapto acids and amino acids in the initial sample and the fraction of the sample that is converted to products that contain amide bonds. These results are consistent with our observation that tg evaporates over time, and hence starting with greater amounts of tg, even with the same tg to Ala ratio, allows more tg to remain in the reaction tube as the sample reaches the dry state.

In addition to Supplementary Figures 1-2, the following text has been added to the Results section:

“Preliminary studies indicated oligomer formation with greater yields when the starting mixture contained an excess of mercaptoacid over amino acid (Supplementary Fig. 1). This is likely partially due to the greater formation of thioester moieties, which enables thioester-amide exchange, consistent with previous investigation with mixtures with varying molar ratios of amino acids and hydroxy acids.³¹ Moreover, excess mercaptoacid allows sustained reactivity due to partial volatility of tg itself during the dry-heating process (Supplementary Fig. 2). We therefore employed a 5:1 mercaptoacid/amino acid molar ratio in subsequent reactions.”

We also added the following text to the Methods:

“For dry-down reactions involving varying amounts of tg and Ala at a 1:1 molar ratio, tg and L-Ala were dried at varying amounts (from 10 μ mol to 250 μ mol) at 65 °C for seven days. Samples were rehydrated with appropriate volumes to give 100 mM Ala concentration (referring to the starting Ala concentration).”

Regarding the comment about the product distribution and potential oxidation products, please see our response to comment #2.

Reviewer 1, Comment 2. *The analysis is qualitative, and products are assumed based on relative binding to C18 silica. The authors only address “combined yields” for total observed products in comparison to the residual amino acid. The authors also do not provide data on the hydrolysis of the reaction products, which would remove all suspected thioesters shown in Supplementary Fig. 2. The authors also do not reduce the disulphide products after the reaction, which would remove all suspected disulphides, shown in Supplementary Fig. 2. If oxidation occurs, how do the authors account for this?*

Response: To address this comment we verified the structures and quantified the yields of four product species that are of particular interest (Figure 2c). To this end, we generated calibration curves for HPLC peaks using four authentic standards: c(tgA), tgA, tgAA, and tgAAA, which includes two new standards, c(tgA) and tgAAA, that were synthesized during manuscript revision. Using these standard curves, and by the co-injection of each authentic standard with reaction product mixtures, we were able to verify the identity and yields of specific products based on LC-UV/MS data. In the revision, we added Supplementary Figure 9 (Validation of the formation of tgAAA peptide upon dry-down of tg with Ala), Supplementary Figure 10 (Validation of the formation of c(tgA) upon dry-down of tg with Ala), and Supplementary Figure 11 (Quantification of products from dry-down reactions using calibration curves of synthesized standards.)

As seen from our quantitative results (Supplementary Figure 11), 45% of Ala is converted into tgA, 3% of Ala is converted into c(tgA), and 6% of Ala is converted into tgAA, and ~0.2% of Ala is converted into tgAAA. Since we know that 83% of Ala has reacted based on our quantitative ¹H-NMR analysis (Figure 4), we can conclude that ~29% of Ala has converted into other types of oligomers.

We have added the following text to the Results:

“A tgAAA oligomer was also identified via comparison to an authentic standard, although tgAAA is in relatively low abundance (Supplementary Fig. 9). LC-MS also indicated a species with the mass of tgA minus a water molecule, at retention time 8.5 min. We hypothesized that this product is the cyclized form of tgA, i.e. (S)-3-methylthiazine-2,5-dione. We confirmed this hypothesis by synthesizing the (S)-3-methylthiazine-2,5-dione standard and verified that the standard is equivalent to the species with 8.5 min retention time (Supplementary Fig. 10).

We quantified various products from dry-down reactions using calibration curves of four authentic standards: c(tgA), tgA, tgAA, tgAAA (Supplementary Figure 11). Our results indicate that 45% of Ala is converted into tgA, 3% into c(tgA), and 6% into tgAA.”

Regarding the concern expressed by Reviewer 1 about oxidation, we have worked in anoxic conditions and observe minimal oxidation, based on MS analysis, in the form of disulfide bond formation (Fig. 2a). To test for potential oxidation, we reduced the dried mixture with TCEP and showed that disulfide bond formation is minimal under our conditions (new Supplemental Figure 4). As background, while we try to minimize oxidation using a Coy Chamber, residual oxygen is possible when the reaction is heated. In our anaerobic chamber the oxygen levels are typically stable at ~20-30 ppm but can fluctuate between 30-100 ppm for very short periods of time.

As seen by our results presented in Supplementary Figure 4, only minimal oxidation occurs upon dry-heating in our anaerobic chamber. In addition to adding this new SI figure, we added the following text to Results:

“We verified that oxidation was minimal under our anaerobic reaction conditions (Supplementary Fig. 4).”

We added the following text to Methods:

“Reduction with tris(2-carboxyethyl)phosphine

To analyze for oxidation levels, we incubated a dried mixture of tg and Ala with 500 mM tris(2-carboxyethyl)phosphine (TCEP) for 1 hr at RT. The resulting products were analyzed by hydrophobicity-based separation using C18-HPLC.”

In response to Reviewer 1’s suggestion to use hydrolysis reactions to selectively degrade thioesters, we incubated in water three days at 65°C, under anoxic conditions, a sample of tg and Ala that had previously been held in the dry state for seven days at 65 °C. This aqueous incubation resulted in hydrolysis of a few product species (new Supplemental Figure 14). As seen by our results shown in Supplementary Figure 14, some product oligomers were susceptible to hydrolysis in water at 65 °C, further supporting the existence of thioester bonds. In addition to adding this new SI figure, we added the following text to Results:

“Incubation in water of a previously dried mixture of tg and Ala for 3 days at 65 °C under anoxia led to partial degradation of various oligomers, further supporting the existence of thioester bonds among the dry-down products (Supplementary Fig. 14).”

We added the following text to Methods:

“Hydrolysis assay

We incubated a previously dried mixture of tg and Ala in water for 3 days at 65°C (unbuffered, pH~3.5) under anoxic conditions. Samples were analyzed by hydrophobicity-based separation using C18-HPLC before and after incubation in water.”

Reviewer 1, Comment 3. *Miller has already describe the effect reported here. In 1995 (Nature 373, 683-685 (1995)), Miller reported dry heating amino acids with thiols gives amides and peptides, “at 40 °C the lowest temperature used so far in which amide bonds are formed”. Miller’s work was also presented in the context of the origins of life. Miller’s paper is cited in a reference string in the introduction, but its outcomes are not explained.*

Response: We cite the Miller 1995 Nature paper in our original and revised manuscript. However, we disagree with Reviewer 1. This paper by Miller does not “describe the effect reported here [in our manuscript]”, nor does any other paper to the best of our knowledge. In the cited paper by Miller and coworkers a high energy chemical species was used to drive their reaction, i.e., the chemical potential of the pantoyl lactone. We do not use any high-energy species, and show formation of thioester and amide bonds in peptide-like molecules by simply drying and heating mercapto acids and amino acids. Moreover, the amines in Miller’s system were on cysteamine or β -alanine, which have very different nucleophilicity compared to alpha amino acids used in our study. For instance, Liu and Orgel showed that β -glutamic acid polymerizes more efficiently than its alpha analog in the presence of activating agent 1-ethyl-3-(3-dimethylaminopropyl)carbodiimide (EDAC) (Liu, R. & Orgel, *J. Am. Chem. Soc.* 119, 4791-4792 (1997)). Furthermore, the formation of amide bonds through thioester-amide exchange, the main point of our manuscript, is not mentioned nor implied in Miller’s work (it was proposed by Miller that β -alanine will ring-open the lactone through ester-amide exchange). Last and most importantly, we show formation

of heterogenous peptide-like molecules, whereas Miller showed the formation of a molecule that resembles Coenzyme-A precursor and not in the context of formation of peptide oligomers.

Reviewer 1, Comment 4. *The title of this paper should specify thiopeptides not peptides.*

Response: The major products of our reactions are SH-terminated peptides (i.e., short peptides with a terminal SH instead of NH₂) and thiopeptides. To accommodate the reviewer's comment, we changed the title of the paper to "Thioesters Provide a Plausible Prebiotic Path to Proto-Peptides".

Reviewer 1, Comment 5. *Freckel-Pinter et al. state: "We report robust, plausible prebiotic reactions of mercaptoacids with amino acids that result in the formation of peptides and thiopeptides, which contain both peptide and thioester bonds." What is meant by "robust", how is this quantified? Are any pure peptide (only alanine products) formed or only thiopeptide (tga + alanine)? Do all products contain tga (thiol or disulfide)? What fraction of new bonds are peptide (alanine-alanine) and what fraction are thiopeptide (tga-alanine or alanine-tga)? What fraction of amides are AA-amides and what fraction are tga-amides? What fraction of new bonds are thioester vs amide bonds? This could be quantified.*

Response: By "robust" we meant that proto-peptides are formed over a wide range of temperature and pH conditions (albeit the best yields are at higher temperatures and acidic pH). To the best of our knowledge, we report the most robust model prebiotic pathway for peptide formation ever reported – particularly without the use of chemical activation.

Reviewer 1 is correct that there is a challenge describing the peptide-like products in this paper, as the oligomers that will contain only peptide bonds are terminated with an SH instead of NH₂ (we can term them 'SH-peptides'), and other oligomers contain both amides and thioesters (i.e. thiopeptides). To remove this confusion, we speak of the formation of oligomers with "peptide bonds" rather than the formation of "peptides," when appropriate.

We did not detect pure peptides (e.g. Ala-Ala) in our reactions in quantifiable yields. All products have at least a terminal tg, as expected by our mechanism (Fig. 1). As mentioned in response to comment #2, only minimal oxidation is observed under our stringent anoxic conditions. We quantified various products of our dry-down reaction (see response to comment #2).

Reviewer 1, Comment 6. *The authors should demonstrate what they have synthesised and quantify specific products – in a heterogeneous reaction this is important. Demonstrating the product distribution is especially important when the core discussion is focused on coupling amino acids, and suggests tga is a "catalyst".*

Response: We have synthesized new standards and quantified numerous species (see response to comment #2).

Reviewer 1, Comment 7. *Is tga catalytic? What turn over number is observed for tga in these reactions?*

Response: See response to comment #1.

Reviewer 1, Comment 8. *As the work is presented the "tga-amide bonds" (in for example, tga-A) are used to suggest that high "peptide yields" are observed, but this compound is not a peptide. Most of the suggested products (Supplementary Fig 2) don't contain peptide bonds – this is not clear in the manuscript.*

Response: See response to comment #5.

Reviewer 1, Comment 9. *Proposed peptide products, such as the alanine tetramer, are commercial and could easily have been verified by the authors.*

Response: As mentioned in response to comment #5, we are not observing the formation of pure peptides with amino acids alone, in accordance with the mechanism involving pre-formation of a thioester between tg molecules (Fig. 1). This is why we had to synthesize tg-terminated peptides ('SH-peptides') to verify the suspected reaction products. These molecules are not commercially available.

Reviewer 1, Comment 10. *Freckel-Pinter et al. suggest a wide range of pHs work. But the range is restricted and does not include any pH above neutral. It is also not clear that the dry reactions are at neutral pH after drying. How is the pH measured or calculated in the dry reactions? What happens at higher pH? How do the authors verify that under the dry conditions the "pH" of the reaction is not acidic. Removing water (volatiles) during drying will significantly shift the pH. For example, Miller (see Nature 373, 683-685 (1995)) reports that solutions of amino acids and thiols at initial pH 7, following drying dropped to pH 4.*

Response: The pH was verified with litmus paper before and after the reaction. Imidazole is not volatile and changes in the pH following dry-heating process were not observed. In the dried phase pH is not a well-defined parameter in any study. We measure the pH in the beginning of a reaction and upon resuspension with water at the end of a reaction.

In response to Reviewer 1's comment about basic pH, we carried out reactions with samples that were made basic by the addition of NaOH and found that some amide bond formation is still observed, although at a rate that is about an order of magnitude less than at acidic pH (using the production of tgA for comparison). The results are shown in the revised manuscript as Supplementary Figure 43. Similar to reactions with imidazole, the pH was verified with litmus paper before and after the reaction. NaOH is not volatile and changes in the pH were not observed following dry-heating process. In addition to adding this new data to the SI, we added the following text to Results:

"To test for thiopeptide formation at basic pH, we carried out dry-down reactions containing mixtures of tg and Ala in which the pH was adjusted using NaOH (Supplementary Fig. 43). Although yields were low, we still observed some product formation at pH 9.0 and even at pH 12.5 (Supplementary Fig. 43)."

We added the following text to Methods:

"For variable pH dry-down experiment, samples were dried under different pH conditions by the addition of different number of equivalents of imidazole or NaOH. For Imidazole-containing reactions, the initial reaction pH was adjusted to 3.5, 5.5, 6.5, or 7.0. One, 5, 10, or 20 equivalents of imidazole were added with respect to the amount of amino acid to adjust the pH. For NaOH-containing reactions, the initial reaction pH was adjusted to 3.5, 9.0, or 12.5. Two, 5, or 10 equivalents of NaOH were added with respect to the amount of amino acid to adjust the pH."

Regarding the comment about Miller's work, Miller used lactone, which is expected to quickly hydrolyze to a hydroxy acid at basic pH, which will result in lowering the sample pH. This is not applicable to our system as we started with molecules in a carboxylic acid/carboxylate form.

Reviewer 1, Comment 11. *The suggested important distinction between this work and the previous work (with esters rather than thioesters) is that the new work is no longer restricted to "acidic conditions".*

This needs to be more rigorously proven. The authors also need to specify the pH range that the previous ester mediated synthesis was restricted to.

Response: For the first part of this comment, see response to comment 10. For the second part of this comment, we reported in the original paper:

“Both depsipeptide and thiodepsipeptide polymerization occurs under acidic conditions, but in contrast to thiodepsipeptides, no depsipeptide products are formed at pH 7 (Supplementary Fig. 48). The activation energy for formation of thiodepsipeptides is lower than for depsipeptides. While products were detected in dry-down reactions containing mixtures of tg and Ala even at temperatures as low as 25 °C (**Fig. 5b**), no products were observed under these conditions for dry-down reactions containing the equivalent hydroxy acid and amino acid mixture (Supplementary Fig. 49). In contrast to thiodepsipeptide formation, which proceeded to some extent in water (Supplementary Fig. 44), incubation of a mixture of glycolic acid (glc, the hydroxy acid analog of tg) with Ala at 65 °C for one week did not produce oligomers (Supplementary Fig. 50). The extent of conversion of a given amino acid into products is higher when dried-down with mercaptoacids compared to when dried-down with hydroxy acids. Specifically, 83% of Ala was converted into products when dried-down with tg, whereas only 64% of Ala was converted into products with glc (**Fig. 4**, Supplementary Fig. 51).”

Reviewer 1, Comment 12. *In the “dry” samples, with 5 eq. tga, what is the bulk material? Powder or liquid? Is the reaction being run in liquid tga?*

Response: The sample is ‘waxy’/gel-like. See image inserted below:

Reviewer 1, Comment 13. *Freckel-Pinter et al. suggest their reaction is the “most robust one-pot pathway for peptide formation ever reported”. This comment is totally unjustified. How have the authors quantified the robustness of their reaction against other reported peptide syntheses. What even makes this process “one-pot”? What does that mean? They have to separately synthesise and isolate tga and alanine, and then mix them, why is this “one-pot”. Again, what is robust here – the authors keep repeating this term, but it is meaningless.*

Response: In response to the comment about robustness and comparison of our reactions against other reported peptide syntheses, please see our responses to comments #5 and #11.

By “one pot” we mean that our synthesis occurs upon mixing of the two building blocks in one reactor (mercaptoacid and amino acid) with no preparation of an activating agent, no pre-activation of the monomers, and no purification steps in between, as common in various other reported prebiotic reactions that involve purifications by chromatography. We agree with the reviewer that no reaction is really “one-pot” synthesis because there was an original source of the reactants. The use of this term was to differentiate this work from work of others that required isolation/purification of some products, whereas our system is approaching a more one pot reaction where no purification is needed. We have changed in the Abstract “Our results offer the most robust one-pot pathway for peptide formation ever reported” to “Our results offer a robust one-pot prebiotically-plausible pathway for proto-peptide formation”.

Reviewer 1 Comment 14. *The reaction is certainly not robust to something as simple as the “wrong” amount of amino acid. It is shut down by more amino acid. This is not a robust reaction.*

Response: Regarding the ratio of mercaptoacid to amino acid, we now show that the excess amino acid does not inhibit the reaction. See our detailed response to comment #1. Regarding use of the term “robust”, again, we changed in the Abstract “Our results offer the most robust one-pot pathway for peptide formation ever reported” to “Our results offer a robust one-pot prebiotically-plausible pathway for proto-peptide formation”.

Reviewer 1, Comment 15. *Freckel-Pinter et al. should cite Foden et al. Science, 2020, 370, 6518, 865-869, that demonstrates tga, and other “simple thiols such as coenzyme M (79%) (fig. 85)”, catalyze peptide formation in water at neutral pH from nitrile and amine reactions. This work was also presented in the context of the origins of life.*

Response: The paper was already cited in our original manuscript as reference 15. This is a beautiful work, but distinct in important ways from the work we present. The mechanism of peptidyl amidine formation shown in Foden *et al.* is practically irreversible, whereas dynamic shuffling is possible with our thioester approach to prebiotic peptide synthesis. To elaborate on this work, we added the following sentence to the Introduction: “Moreover, acetylcysteine or derivative thiols were shown to catalyze peptidyl amidine formation in water when combined with α -aminonitriles.”¹⁵

Reviewer 1, Comment 16. *Freckel-Pinter et al. suggest they observed high conversion to “oligomers”, this is exaggerated by including the single reaction of tga and amino acid in the yield of “oligomers”. This is not an oligomer, it is not even a dimer, it is the product of a reaction between two different compound types. The authors could easily deconvolute the yield of this single major reaction product from the “oligomer” yield. The “oligomers” also appear to incorporate significant numbers of disulphides, these compounds are likely exaggerating the “oligomer” yield which is presented as amide or peptide or thiopeptide in the paper.*

Response: The reviewer is using a strict definition of dimer. tgA is a dimer, or more specifically a heterodimer (hence the smallest oligomer), since it forms upon condensation-dehydration between two building blocks (tg and Ala). Regarding quantification of specific product peaks, see our response to comment #2. Regarding “significant numbers of disulphides”, see our response to comment #2 (only minimal oxidation is observed under our anoxic conditions).

Reviewer 1, comment 17. *What happens if “longer” oligomer products (or suspected products) are subjected to the conditions of the reaction? Are they stable? Most of the products shown in Supplementary Fig. 2 are expected to be significantly less stable than peptides.*

Response: We agree with the reviewer that products of these reactions will exhibit varying hydrolytic stability, depending on the nature of their backbone. Specifically, oligomers that contain only amide bonds will exhibit much longer hydrolytic stability (i.e. persist longer in solution) compared to thioester-containing compounds, the latter are much more labile (see response to comment #2). Following the reviewer’s suggestion, we took various products of our reactions and for simplicity purposes we focused on reactions that contain dimers (which are major products of our reactions). To that end we synthesized two additional compounds: an Atg standard that contains a thioester bond and the thiazinedione c(tgA) that is the ring form of tgA. We then performed additional reactions in which we used either tgA standard (containing an amide bond), Atg, or the thiazinedione c(tgA) in the presence or absence of either tg or

Ala, which proved to be informative. For instance, our results indicated that the thiazinedione (confirmed to be a product in our reactions via the newly synthesized standard) can undergo fast ring-opening polymerization with either tg or Ala (Supplementary Fig. 17).

We added the following text to the Results section:

“For a more detailed mechanistic investigation using plausible reaction intermediates, we performed reactions in which we used either the tgA standard (containing an amide bond), an Atg standard (containing a thioester bond), or the thiazinedione c(tgA) in the presence or absence of either tg or Ala. We suspected that c(tgA) can undergo ring-opening polymerization with tg or Ala (Supplementary Fig. 16).⁴²⁻⁴⁴ To test this hypothesis, we dried c(tgA) in the absence or presence of tg or Ala. Indeed, we observed fast formation of products following dry-down that would be expected to result from ring-opening polymerization, such as tgAA (Supplementary Fig. 17). A notable thioester hydrolysis product was observed as well, as some tgA also occurred following drying of c(tgA) at 65 °C (Supplementary Fig. 17). On the other hand, reactions involving the Atg standard showed fast thioester hydrolysis followed by formation of tgA from free tg and Ala (Supplementary Fig. 18), whereas reactions of tgA showed formation of species that indicated some amide hydrolysis of the tgA standard (e.g. the tgAA product was observed, Supplementary Fig. 19).”

We added the following text in Methods:

“For dry-down reactions involving the tgA, Atg, or c(tgA) standards, the standards were dry-heated at 65 °C for up to seven days in the absence or presence of either alanine or tg at a 1:10 molar ratio (in favor of either tg or alanine).”

Reviewer 1, Comment 18. Fig. 5. What is “tgA” vs “tgA-H₂O”? What is the structure of tgA-H₂O?

Response: tgA is a tg linked through an amide to Ala. tgA-H₂O is an ionic species observed in the mass spectrum with m/z value corresponding to tgA minus 18 amu (the mol. wt. of H₂O). In our original manuscript we assigned tgA-H₂O to the closed ring form of tgA (i.e. a thiazinedione, herein termed c(tgA)). The structures of tgA and c(tgA) are:

To confirm that the species with the mass of tgA-H₂O is indeed the thiazinedione, we synthesized a (S)-3-methylthiazine-2,5-dione as an authentic standard and verified that it is the product indeed (see new Supplementary Figure 10 and response to comment #2).

As mentioned, we added the following text in Results:

“LC-MS also indicated a species with the mass of tgA minus a water molecule, at retention time 8.5 min. We hypothesized that this product is the cyclized form of tgA, i.e. (S)-3-methylthiazine-2,5-dione. We confirmed this hypothesis by synthesizing the (S)-3-methylthiazine-2,5-dione standard and verified that the standard is equivalent to the species with 8.5 min retention time (Supplementary Fig. 10).”

We have changed the labelling of “tgA-H₂O” to c(tgA) in Figures 2 and 5.

Reviewer 1, Comment 19. Fig. 5. What are the major peaks at 3 mins and 15 mins at 25 – 50 °C? Why do both disappear at 65 °C? If this is loss of a volatile – how are yields accurately quantified?

Response: The peak at retention time 3 min is the tg monomer and the peak at ~14 min is the tg dimer that contains a thioester. These peaks are labeled in Fig. 5 of the manuscript. At 65°C the tg is more reactive

compared to lower temperatures and hence other species are observed. Moreover, tg is partially volatile (see our response to comment #1). The yields are calculated based on Ala, which is not volatile.

Reviewer 1, Comment 20. *Freckel-Pinter et al. state: “Kinetic analysis shows gradual oligomer formation by tga and Ala for up to seven days in the dry state (Supplementary Fig. 7).” Why is this a kinetic analysis? It is just a time course. Rates could not be extracted from the data provided. The authors also state, with no evidence, that gradual oligomer formation occurs? How do they prove this? How do they quantify the total content of material – this is not clear.*

Response: The gradual consumption of monomers (tg and Ala) and growth of products, such as tgA and tgAA, shown in Supplementary Fig. 15 was considered evidence of gradual oligomer formation. We agree that it is a qualitative analysis and not quantitative. We changed “kinetic analysis” to “time course analysis”.

Reviewer 1, Comment 21. *Over seven days the author show that tgaA grows significantly. But it is not clear that the other product are not due to, for example, oxidation? Why can't the data acquired in Supplementary Figure 10 and Figure 13, and HPLC of the corresponding samples (e.g. Fig. 5) be used to quantify the amount of tgaA? The same could be done for tgaAA (see Figure 6). This data could then be used to assess the absolute conversion and relative yields.*

Response: Done – please see response to comments #2 and #5.

Reviewer 1, Comment 22. *Freckel-Pinter et al. report LCMS. The authors should be able to calculate how much peptide and how much thiopeptide of each composition is present in the mixture. Why don't they?*

Response: Our MS analysis offers qualitative analysis and not quantitative since different molecules have different ionization propensity. Hence, based on MS we can only verify the existence of molecular species, but not quantify amounts. Our UV-monitored HPLC and quantitative NMR analyses were used as complementary quantitative methods (see response to comment #2).

Reviewer 1, Comment 23. *Freckel-Pinter et al. interpret NMR peaks as “oligomers”, which they then infer to be “peptides”, when they have not quantify the content of tgaA. The simplicity suggests the product is largely one compound. The NMRs all lack the complexity expected for significant amounts of oligomer products.*

Response: For quantification, please see our response to comment #2.

Reviewer 1, Comment 24. *Why do the authors show 1H-1H NMR, that simply verifies the Ala connectivity of the starting material. Why don't the authors provide 1H-13C NMR that could be used to verify the reactions discussed.*

Response: We are confident in our assignment of products reported our study. These assignments have been verified by a variety of analytical methods, including LC-MS, FTIR, UV-HPLC, ¹H-¹H NMR, together with synthesized authentic standards that were co-injected with products to confirm column retention times of specific molecular species.

Reviewer 1, Comment 25. *What are the 1H-1H signals between peaks at 3.5 ppm - 4.2 ppm? These are*

not identified. They are not alanine peaks, what are they? They are relatively major products, likely second major product, after tgaA.

Response: The free Ala has a chemical shift of 3.95 ppm, and the black circle shows correlation between the α -proton and methylene protons of non-reacted Ala. The other small peaks are probably associated with dried tg. Below is ^1H NMR spectrum of a dried tg control following 1 week of heating at 65 °C in the dry state (resuspended in deuterated MeCN), showing peaks between 3.6 and 4.2:

Reviewer 1, Comment 26. *Supplementary Fig 19 is mislabelled, both traces are labelled as the control.*

Response: We thank the reviewer for pointing out this error. We have labeled the bottom trace “tg + L-Phe (5:1)”.

Reviewer 1, Comment 27. *Supplementary Fig 20: what is the large signal at 2.1 ppm after dry down? It is a major component but does not apparently relate to tga or cysteine. What are the large signals in the starting materials at 3.8 ppm and 3.6 ppm? Cysteine is a thiol-acid. Why is tga needed in this reactions?*

Response: We believe that the reviewer refers to original Supplementary Figure 24 (Now numbered as 36). The large signal is likely the degradation products of cysteine monomer. Regarding the use of tg instead of cysteine, subjecting cysteine by itself to a dried state does not lead to appreciable product formation (Supplementaty Figure 35). Notably, in previous work with hydroxy acid-mediated peptide bond formation it was found that alpha hydroxy acids were particularly effective in catalyzing peptide bond formation and that serine, the alcohol analog of cysteine, was not. This work was largely motivated by our hypothesis that substituting hydroxy acids with mercapto acids would lead to peptide bond formation under a wider range of conditions, which is why we have focused on tg (an alpha-mercapto acid).

Reviewer 1, Comment 28. *Supplementary Fig 25: why is the spectrum cut at 3.0 ppm? Is the unexplained peak from Fig 20 here too? As above, what are the large signals in the starting materials at 3.8 ppm and 3.6 ppm? The authors state 66% of Phe has converted into oligomer, but what fraction of this is Phe-Phe peptide and what fraction is just tga-Phe?*

Response: We replaced the spectrum that was originally Supplementary Fig 25 with the full spectrum (now new Supplementary Figure 37). The unexplained peak from dried cysteine is not seen here. In the spectrum of dried tg+Phe there is a peak close to this region, but this peak is a resonance of deuterated MeCN that was added to D₂O to properly dissolve the reaction products (also shown in the dried tg sample resuspended with deuterated MeCN in response to comment #25). No deuterated MeCN was used to dissolve the dried samples containing cysteine. Regarding the signals in 3.64 and 3.78 ppm in the starting material, these are resonances of (tg)₂ (a tg dimer with a thioester bond), which is present in the tg stock. We added the two diagonal lines above the tg monomer protons to indicate that the signal is cut to zoom into the Phe protons.

Regarding the question about what fraction of oligomers are ‘Phe-Phe peptides’, see our response to comment #5.

Reviewer 1, Comment 29. *Why is imidazole used in the “pH” range? What happens if the pH is adjusted with NaOH rather than imidazole? How do the authors ensure the pH is not changed during dry down. Are the pH values measured or calculated? What do the authors mean by “referring to the number of eq. of acid”? Do they mean 20 eq. for tga and 20 eq. for Ala? If so this is 40 eq. Are the pH values of these reactions checked before and after drying? If so they should be reported, if not, why not?*

Response: Regarding the choice of imidazole, it was chosen due to its pKa and since it enables a gel-like state during the dry-heating process, which will enable molecule diffusion and reactivity. Regarding the reviewer’s comment about NaOH, see response to comment #10. Please also refer to response to comment #10 for the other points raised regarding pH measurements.

We added the following text to Results: “pH was measured before and after the reactions. The pH did not fluctuate more than 0.5 pH units after the reaction.”

Regarding the question about “referring to the number of eq. of acid”, in the Results section we specified the following: “To test this possibility, we carried out dry-down reactions containing mixtures of tg and Ala in which the initial reaction pH was adjusted to 3.5, 5.5, 6.5, or 7.0. One, 5, 10, or 20 equivalents of imidazole, with respect to the amount of amino acid, were added to adjust the pH.”

Following the reviewer’s comment, we added this clarifying sentence in the Methods section: “For variable pH dry-down experiment, samples were dried under different pH conditions by the addition of different number of equivalents of imidazole or NaOH. For Imidazole-containing reactions, the initial reaction pH was adjusted to 3.5, 5.5, 6.5, or 7.0. One, 5, 10, or 20 equivalents of imidazole were added with respect to the amount of amino acid to adjust the pH.”

Reviewer 1, Comment 30. *Supplementary Figure 31: What concentration are the compounds? The authors state “Various product peaks are evident even in the wet phase, some of which contain amide bonds (such as tgaA).” What is meant by “wet phase”? What is the concentration? Is this a solution? The authors don’t provide a “pH” for the reaction in “wet phase”, they need to give this pH. Have the authors demonstrated that this “wet” reaction yields peptides (i.e. AA bonds)? This should be clarified in*

the text. The only labelled peak doesn't contain peptide, only tgaA. What are the other "peaks"? What are the peaks at 3 mins, 12 mins, and 14 mins?

Response: We state in the Methods section: "For the pH-dependent experiment in solution, samples were incubated in water (pH~3.0, 55.55 mM Ala and 277.75 mM tg) or in the presence of 20eq Na₂HPO₄ (pH~7.0, referring to the amount of the amino acid, 38.46mM Ala and 192.30mM tg)."

To clarify, we added the concentrations in the legend of Supplementary Figure 31 (Now numbered as Supplementary Figure 44) as well.

Regarding the question about the wet phase, we were referring to reactions that occur in aqueous solution and that were not dried-down. We added a clarification in the legend of Supplementary Figure 44: "Various product peaks are evident even in the wet phase (reactions in aqueous solution that were not dried-down)".

The reaction pH was already mentioned in the legend of Supplementary Figure 31 (Now numbered as Supplementary Figure 44) -pH~3.0 in unbuffered acidic reactions or at pH 7.0 (using a phosphate buffer), as well as in the Methods section.

The reaction produces tgA, which is a heteropeptide (contains an amide bond, see our response to comment #5). We labelled the peaks in retention times 3 (tg), 12 [(tg)₂ S-S], and 14 ((tg)₂ with a thioester bond).

Reviewer 1 Comment 31. *Low "yields" from wet-dry cycles may indicate the instability of the products, casting further doubts on the claim of robustness.*

Response: We anticipated lower product yield from these wet-dry cycled reactions because the monomers spend less time in the dry state (compared to continuous dry-state reactions of the same duration). Wet-dry cycling will oscillate the system of proto-peptides between a dry state that favors condensation and a wet state that favors hydrolysis. We view the reversibility of thioesters, and peptide bonds, to be an advantage rather than as a disadvantage, since this reversibility allows for fast dynamic chemistry, e.g. monomer shuffling, which has the potential to expand the sequence space explored by proto-polypeptides, and could greatly accelerate the discovery of functional sequences during the early prebiotic chemical evolution.

Reviewer #2 (Remarks to the Author):

Frenkel-Pinter et al. report on the reaction of amino acids with thiols. The authors show that by drying these compounds together, two key functional groups of metabolism: thioesters and peptide bonds can be formed. Furthermore, the authors show that the thiopeptides also form in water, which is surprising and extends the relevance of the work. Finally, the authors are able to compare their results with previous results from their groups, highlighting the key difference of sulfur in a thiol as a nucleophile in comparison to a hydroxyl group with the formation of peptides: peptides are unable to form at neutral pH, at low temp., and not in water, which is in contrast to the thiol derived results presented. Altogether these data are a significant advance.

The writing is clear and the results are fairly straightforward. The results are original and will be important not only for work in the origins of life field but also as written by the authors, more broadly for example in the study of combinatorial libraries, and even possibly materials chemistry. There are no major flaws in the interpretations or claims made by the authors that I see, and the work is reported in sufficient detail.

Response: We thank the reviewer for their positive assessment of our work and for constructive suggestions, which have improved our manuscript.

I have a few small comments that I hope might be of use for the authors to improve the article.

Reviewer 2, Comment 1. *Figure 1) I'd like to suggest the authors add some more material here. Specifically, how about adding a reaction showing the formation of the aminoacid-thioester, which is presumably needed to form dipeptides such as the tg-AA? In the current form, I may lack the needed chemical imagination but I stumbled on how to go from the tg-amide to an amino acid dimer. I think adding at least this one more reaction would be useful, and the authors might consider adding others, which would elevate the work and make it more interpretable.*

Response: The reviewer is right that thioester intermediates are required for formation of amide bond through thioester-amide exchange. To make this point clear, we added a scheme in the SI suggesting two possible routes for the formation of tgAA (see new Supplementary Figure 12).

In the Results section we wrote: “Products containing sequential amide bonds, such as the tgAA molecule, can form through thioester intermediates, for instance between tg and tgA, as illustrated in Supplementary Fig. 12.”

Following the reviewer's suggestion to add a few more reactions, we synthesized two additional compounds: an Atg standard that contains a thioester bond and the thiazinedione c(tgA) that is the ring form of tgA. We then performed additional reactions in which we used either tgA standard (containing an amide bond), Atg, or the thiazinedione c(tgA) in the presence or absence of either tg or Ala, which proved to be informative. For instance, our results indicated that the thiazinedione (confirmed to be a product in our reactions via the newly synthesized standard) can undergo fast ring-opening polymerization with either tg or Ala (Supplementary Fig. 17).

We added the following text to the Results section:

“For a more detailed mechanistic investigation using plausible reaction intermediates, we performed reactions in which we used either the tgA standard (containing an amide bond), an Atg standard (containing a thioester bond), or the thiazindione c(tgA) in the presence or absence of either tg or Ala. We suspected that c(tgA) can undergo ring-opening polymerization with tg or Ala (Supplementary Fig. 16).⁴²⁻

⁴⁴ To test this hypothesis, we dried c(tgA) in the absence or presence of tg or Ala. Indeed, we observed

fast formation of products following dry-down that would be expected to result from ring-opening polymerization, such as tgAA (Supplementary Fig. 17). A notable thioester hydrolysis product was observed as well, as some tgA also occurred following drying of c(tgA) at 65 °C (Supplementary Fig. 17). On the other hand, reactions involving the Atg standard showed fast thioester hydrolysis followed by formation of tgA from free tg and Ala (Supplementary Fig. 18), whereas reactions of tgA showed formation of species that indicated some amide hydrolysis of the tgA standard (e.g. the tgAA product was observed, Supplementary Fig. 19).”

We added the following text in Methods:

“For dry-down reactions involving the tgA, Atg, or c(tgA) standards, the standards were dry-heated at 65 °C for up to seven days in the absence or presence of either alanine or tg at a 1:10 molar ratio (in favor of either tg or alanine).”

Reviewer 2, Comment 2. *Figure 2) (and elsewhere) in the figure structures of fully protonated tg is drawn, and also carboxy-protonated alanine. Are they the correct forms under negative ionization mode? please confirm.*

Response: We redrew the structures in Figure 1 and Figure 4 such that their predominant forms in the dry-down reactions under acidic conditions (pH ~3.0), regardless of their ionization in negative or positive mode ESI analysis, will be represented and added this in figure legend. The pKa of thioglycolic acid is 3.8, so we kept the carboxylic acid rather than the carboxylate form. For Ala we fixed the NH₂ to NH₃⁺ and changed from a carboxylic acid to a carboxylate (pKa=2.34). We kept the structures in Figure 2 (MS data) as neutral since their ionization will depend on the mode (negative or positive).

Reviewer 2, Comment 3. *Figure 2C) Whereas thioesters show an absorbance feature at ~230nm, peptides show an absorbance at 280nm. It could be useful to show two traces here, one at each wavelength.... just an idea; up to the authors.*

Response: We thank the reviewer for this excellent suggestion. Indeed, the UV-VIS spectra of amide-containing molecules looks very different from that of a compound containing a thioester, the latter has a unique positive peak at ~230-240 nm. We analyzed the full UV-VIS spectrum of our HPLC runs following dry-heating reactions of tg and Ala and were able to detect peaks containing the typical thioester maxima at ~235 nm, and added this data as a support for the formation of thioester-containing compounds in our dry-down mixtures (see new Supplementary Figure 13).

In the Results section we have added:

” The existence of thioester-containing compounds was verified via collection of full UV-VIS spectra during HPLC analysis, which demonstrated the existence of various peaks with the typical thioester absorbance peak at ~235 nm (Supplementary Fig. 13).^{40,41}”

Reviewer 2, Comment 4. *SI figures 5 and 6. It might be best to stack the traces on top of each other, since they align so closely that it's hard to see each one.*

Response: Following the reviewer’s request, we have added the stacks as inserts to panel A of these supplementary Figures.

Reviewer 2, comment 5. *Supplemental figure 7. Can the authors identify the peaks? Additionally, in these traces (as well as elsewhere) some peaks increase and some decrease during the course. Can the authors comment on that?*

Response: Peaks were labelled in Supplementary Figure 7 (currently renumbered as Supplementary Figure 15). We expanded the sentence describing the kinetic analysis in the Results section as follows: “Time course analysis of dry-state reactions shows gradual oligomer formation by tg and Ala for up to seven days with an initial rise in the thioester-containing tg dimer that is gradually consumed by conversion to other products (Supplementary Fig. 15).”

REVIEWER COMMENTS

Reviewer #1 (Remarks to the Author):

The authors have made significant improvements to their manuscript.

This is an interesting paper, and one that should be published - but there are still changes that they must make for this manuscript to fairly reflect the results that they have acquired.

General points of concern, that must be addressed before publication:

a) the presentation of compounds that “can’t be observed” as products of the reaction this is misleading (e.g. in Sup. Fig. 3, such as (Ala)₄);

b) efficacy and yield of the reaction when Ala > tg;

c) the suggestion that tg is a catalyst rather than a reagent;

d) stability of oligomers where $n > 2$ and degradation via c(tgA) limits the growth of oligomers.

Specifics points:

1) Abstract and Introduction both state: “The condensation of building blocks into oligomers and polymers was an early and important stage in the origins of life.”

This is fine, but obviously speculation and must be written as such, e.g. "It is widely assumed that condensation of building blocks into oligomers and polymers was an early and important stage in the origins of life."

2) Abstract: "Mercaptoacids catalyze thiodepsipeptides and peptide bond formation under ..."

They don't have the evidence to prove this. First, tg is a reagent and incorporated into all of the products, and not proven to be catalytic. Second, all products are thiodepsipeptides.

It would be much more appropriate for the authors to write: "Mercaptoacids and amino acids form thiodepsipeptides under ..."

3) Page 3: The authors write: "We hereby report a much more robust prebiotic system for the formation of peptides and thiodepsipeptides by the reaction of mercaptoacids with amino acids."

They only observed thiodepsipeptides, delete "peptides and".

4) Page 3: The authors write: "While this depsipeptide formation reaction is a promising prebiotically-relevant approach for to peptide bond formation of peptides on the abiotic Earth, it is limited to acidic conditions. ... Compared to reactions involving hydroxy acids as catalysts, mercaptoacids catalyze thiodepsipeptide and peptide bond formation under a wider range of pH conditions, and at much milder temperatures, and at higher water activity."

First, there is no evidence these are catalysts rather than reagents. The authors should change "catalysts" to "reagents". The authors should removed "mercaptoacids catalyze".

Second, delete "and peptide"; the products are all thiodepsipeptides.

Third, it is not possible for readers to compare the thiodepsipeptide and depsipeptide reactions as the reader doesn't know how acidic their depsipeptide synthesis must be to work. Provide the pH required to make the depsipeptide reaction work.

5) Page 3: The authors write: "It is known that thiols can condense with carboxylic acids to form thioesters."

Where is it known? Who reported this? Under what conditions? Appropriate references are required here.

6) Page 4: The authors write: Thiodepsipeptides and peptides form via dry-down reactions of mercaptoacids and amino acids. To test whether thiodepsipeptides form ..."

They only observed thiodepsipeptides, they should delete "and peptides".

7) The authors state in their response to previous comments 1 and 14: The state: "In the revised manuscript we have included data from new experiments that provide additional support for the proposed thioester-amide mechanism and that confirm the presence of amide bonds in the products of these reactions. In the first series of these experiments, the products of constant amino acid concentration (Ala) and varying concentration thiol (tg) reactions demonstrate that increasing tg increases the rate of tgA formation (which contains an amide linkage) (Now included as Supplementary Figure 1). This data shows that increasing the mercapto acid concentration increases the rate of amide bond formation, supporting the role of thiols in catalyzing amide bond formation." And then they state: "Regarding the ratio of mercaptoacid to amino acid, we now show that the excess amino acid does not inhibit the reaction."

The authors statements do not match the data shown.

Increased reagent (tg) correlating with increase yield of products does not provide evidence for "catalysis". The authors should remove any reference to catalysis. They have no evidence for tg catalysis. They are over interpreting their data. What they observed is increase loading of tg gives

more “thiopeptide products” and increased loading of Ala gives less “products”, but as tg is in all products it is a reagent and it cannot be assumed to be a catalyst based on the data provided.

In the chromatograms for 1:1 tg:Ala and for excess Ala, in Sup. Fig 1, there is almost no product – so how can the authors say they “show that the excess amino acid does not inhibit the reaction”? The data presented shows the opposite. Ala inhibition is not necessarily a problem for publication, but the authors must comment on this in the manuscript more explicitly.

They should provide yields for reactions with 1:1 tg:Ala, and excess Ala. This is important. Without these yields how can they say Ala doesn’t inhibit the reaction? I can’t understand why the authors don’t provide a table of yields for the identifiable products for these reactions (i.e. each reaction in Sup. Fig. 1 and Sup. Fig. 2). Why only give a yield at 5:1 tg:Ala? They have developed calibration curves for the major compounds observed to form from Ala, use them. Without this quantification the reader cannot understand how well the reaction works and what the distribution of (major) products is under the different conditions reported.

8) The authors have added two “replicates” of data that had already been presented, i.e. 5:1 tg:Ala and 2.5:1 tg:Ala in Sup. Fig 1. They appear to be different in A and B. Why?

9) The peak marked “Ala” in Sup. Fig. 1B appears to be in Sup. Fig. 1A “control” (that should contain no Ala). Why? Is this contamination.

10) As tg is volatile the ratio of tg:Ala should decrease as the reactions are heated, and therefore lower productivity is expected over time. This should be discussed.

The authors suggest the reaction is “near equilibrium” (“These near-equilibrium reactions are driven by modest environmental changes,”). If this were true then tg loss should drive the “equilibrium” to lower yields over time, as volatile tg is lost. If this is observed the authors should report that in the paper. If it is not observed, then the reaction is not “near equilibrium” as implied, and then the authors should remove discussion of thermodynamic and equilibrium control from the paper.

How do the authors demonstrate that these reactions are “near equilibrium”?

11) Page 4: The authors write: “Direct infusion MS and LC-MS analyses indicated that in dry-down reactions of tg and Ala produced co-oligomers with varying compositions of tg and Ala, and with lengths of co-polymers of tg and Ala were formed (Fig. 2a). Mixtures of tg and Ala produced oligomers up to hexamers, with varying lengths and compositions of tg and Ala.”

There is no need to repeat the same information twice. Delete second instance.

12) Page 6: The authors state: “Incubation in water of a previously dried mixture of tg and Ala for 3 days at 65 °C under anoxia led to partial degradation of various oligomers, further supporting the existence of thioester bonds among the dry-down products (Supplementary Fig. 14).”

Their data in Sup. Fig 14 appears to show that tgA goes down. Why? This is not a thioester.

13) The authors have now synthesised tgAAA and found that it cyclises to 3-methylthiazine-2,5-dione. A nice addition to this paper. But they are observing the thiopeptide undergoes a reaction that degrades the peptide bonds.

The formation of tg-depsipeptides and cyclisation to form thiazine-2,5-diones must be expected to curtail the synthesis of oligomers ($n > 2$), especially with excess tg. This reaction will degrade oligomeric products to yield “tgA dimer”.

This seems to explain why the only major product is tgA, and indicates this may be a poor strategy for peptide growth beyond tgA dimer. This seems to be what the authors results demonstrate.

Furthermore, on this point, the authors were asked (comment 17): What happens if “longer” oligomer products (or suspected products) are subjected to the conditions of the reaction? Are they stable? In their Response: “[They] agree with the reviewer ... Following the reviewer’s suggestion, we

took various products of our reactions and for simplicity purposes we focused on reactions that contain dimers..."

But, although the authors have agreed comment 17 was originally a good question they should address, they then have answered a different question (that misses the point of the original suggestion to look at longer oligomers under the reaction conditions).

They have only looked at the dimer, the shortest possible "oligomer" (assuming tgA is a "dimer" and is therefore the shortest oligomer; I don't subscribe to this as the two parts of tgA are different, but the authors have explained already they do - so, putting the semantics of the prefix "oligo" aside), importantly the "dimer" would not be expected to display the same degradation reactivity as any longer oligomers.

Their observations with tgAAA likely vindicates the original question, and further supports the need to perform the reaction suggested in comment 17. Comment 17 specifically asked what happens when "longer" oligomers are introduced into the reaction, suspecting they would degrade, like tgAAA has been observed to degrade.

There is no doubt that the longer oligomers (and polymers) are presented as the desired products, the authors are clear they believe this is an "important stage in the origins of life", but their results suggest longer oligo/polymers will be degraded by repeated formation of c(tgA) or c(tg)₂. It is very important to address whether the desired products (i.e. oligomers n>2) are stable to the reaction conditions.

The authors should either provide this reaction, originally requested in comment 17, to address the stability of (the desired) oligomeric products (where n>2) or make clear in the discussion within the paper that based on the results they have, they would expect products where n>2 to be disfavoured, due to peptide degradation by tg as is observed for tgAAA.

14) Page 6: The authors imply c(tgA) undergoes a "ring-opening polymerisation". The authors make this sound like the predominant mode of reaction, but this is not supported by the evidence in Sup. Fig. 17.

The major product in Sup. Fig. 17 is hydrolysis, and a very small (unquantified) amount of tgAA is identified too – I see no identification of “polymers” and little evidence they might be there.

Moreover, if it is true, the product of this “polymerisation” would not be well exemplified by “tgAA” in general (as is implied in the text). The addition of A would be an initiation, the polymerisation would then go on to make tgAtgAtgAtgAA type products. In this ring-opening polymerisation the new bonds (other than the single initiation event) are thioesters, not amides. And should be expect to hydrolysis back to tgA.

15) Page 9: Fig. 4A The authors should label the peaks they see in the starting material spectrum at 3.8 and 3.6 ppm.

Could this compound contribute to the yield of tgA? Is it a contaminant in the starting material added to the reaction?

16) The author state in their responses to the reviewer that they use imidazolium salts to adjust the pH of their reaction as it forms fluid gel-like phases upon drying: “Regarding the choice of imidazole, it was chosen due to its pKa and since it enables a gel-like state during the dry-heating process, which will enable molecule diffusion and reactivity.”

This is interesting and potentially important. The authors should communicate this to the reader in the paper.

It also opens the important question of whether the reaction does not work (or is much less effective) if another, less gel-like, buffering salt is used in place of imidazolium? Were other acids not investigated?

Further work on this might be beyond the scope of this paper - but it should be clear to the reader that all these reactions form gel-like or waxy states on drying and this is likely, or at least assumed by the authors to be, an important requirement for this reaction to proceed.

Reviewer #2 (Remarks to the Author):

The manuscript by Frenkel-Pinter et al has been improved by the review process. The realization of thiopeptides through dry down experiments is a natural extension of previous work showing peptide formation, and the occurrence at elevated pH - and also in water - provides the community with a significant advance in our understanding of how peptides can be formed under conditions relevant to the early Earth.

I have a few small final comments that I hope might be useful to the authors.

While re-reading the statements in the abstract and introduction which state that previous work required “activated building blocks or condensing agents” I recalled Huber et alia’s work from 1998 and 2003:

31 JULY 1998 VOL 281 SCIENCE

15 AUGUST 2003 VOL 301 SCIENCE.

In those papers, peptides were formed in a reaction with CO.

Related to this, Leman et al demonstrated high yields of peptides synthesized with COS.

SCIENCE VOL 306 8 OCTOBER 2004

Synlett 2017, 28, 68–72

These results showed at least 2 paths for making peptides in water without “activated building blocks” and while CO and COS might be viewed as a “condensing agent” these are certainly relevant to Archean atmospheres:

<https://www.pnas.org/content/106/35/14784>

To this reviewer, softening the framing of the current work in light of these past observations seems prudent and it would be useful for readers if at least one of the Huber and Leman papers were cited.

Also about the references:

“thioester intermediates enable the synthesis and degradation of peptides, fatty acids, sterols, and porphyrins.7,8 “

Are references 7 and 8 correct here? They don't seem to discuss what is being referred to.

Perhaps an article or book chapter like this would serve the authors intent (though there are of course others)

7.11 - Coenzyme A Biosynthesis and Enzymology

Erick Strauss

<https://doi.org/10.1016/B978-008045382-8.00141-6>

and:

“ long before the great oxidation event that occurred only around 2 billion years ago.45”

Reference 45 may be out of place here. A more appropriate reference would be a recent and excellent review:

Catling and Zahnle, *Sci. Adv.* 2020;6:eaax1420

Also please change “around 2 billion years ago” to “around 2.4 Ga”. 400 million years is a long time.

Supplemental figure 9 and the formation of tgAAA.

These data are very unconvincing and could probably be used to argue that tgAAA was not synthesized in the reaction if I'm understanding it correctly. In panel A, the dry down does not show the expected peak. In panel B) the spike adds only a very small peak to the peak labeled with an arrow. Why is it so small? Can the authors verify this?

The authors write that this was 50µM of tgAAA; what was the spike amount in panel A?

In all of the spiked and standard sample chromatograms shown in the paper, it would be beneficial to indicate the concentration - or better yet - the mol amount injected.

Response to reviewers

Reviewer 1:

The authors have made significant improvements to their manuscript.

Response: We are pleased to learn that the reviewer is happy with the changes made and thank them again for constructive comments.

This is an interesting paper, and one that should be published - but there are still changes that they must make for this manuscript to fairly reflect the results that they have acquired.

General points of concern, that must be addressed before publication:

Reviewer 1, Comment 1: a) the presentation of compounds that “can’t be observed” as products of the reaction this is misleading (e.g. in Sup. Fig. 3, such as (Ala)₄);

Response: We removed structures of “pure” peptides, such as (Ala)₄ in newly revised manuscript. We agree with the reviewer that presentation of such structures might be confusing.

Reviewer 1, Comment 2: b) efficacy and yield of the reaction when Ala > tg;

Response: See response to Comment #11.

Reviewer 1, Comment 3: c) the suggestion that tg is a catalyst rather than a reagent;

Response: In the newly revised manuscript we removed the description of tg as a catalyst.

Reviewer 1, Comment 4: d) stability of oligomers where $n > 2$ and degradation via $c(tgA)$ limits the growth of oligomers.

Response: See response to Comment #14.

Specific points:

Reviewer 1, Comment 5: 1) Abstract and Introduction both state: “The condensation of building blocks into oligomers and polymers was an early and important stage in the origins of life.”

This is fine, but obviously speculation and must be written as such, e.g. “It is widely assumed that condensation of building blocks into oligomers and polymers was an early and important stage in the origins of life.”

Response: we modified the manuscript as requested by the reviewer.

Reviewer 1, Comment 6: 2) Abstract: “Mercaptoacids catalyze thiopeptides and peptide bond formation under ...”

They don't have the evidence to prove this. First, tg is a reagent and incorporated into all of the products, and not proven to be catalytic. Second, all products are thiodepsipeptides. It would be much more appropriate for the authors to write: "Mercaptoacids and amino acids form thiodepsipeptides under ..."

Response: In addition to our response here, please see parallel responses to Reviewer 1, Comment 3, Comment 8, and Comment 11, all of which deal with this same issue – the formal definition of a catalyst.

We have modified the text of the revised manuscript as requested by the reviewer. We have removed description mercaptoacids as catalysts.

Our data show that (i) mercaptoacids are necessary for the formation of peptide bonds under the conditions of our reactions, (ii) the mechanism of peptide bond formation is via thioester intermediates, and (iii) thiols are consumed and then released during the formation of peptide bonds. The formal definition of a catalyst and the description of mercaptoacids as catalysts, or not, does not bear on the fundamental conclusion of our manuscript. Nevertheless, as requested, we have removed that descriptive term.

Reviewer 1, Comment 7: 3) Page 3: The authors write: "We hereby report a much more robust prebiotic system for the formation of peptides and thiodepsipeptides by the reaction of mercaptoacids with amino acids."

They only observed thiodepsipeptides, delete "peptides and".

Response: We modified the manuscript to "HS-peptides" (peptides terminated with a thiol instead of an amine group) instead of "peptides".

Reviewer 1, Comment 8: 4) Page 3: The authors write: "While this depsipeptide formation reaction is a promising prebiotically-relevant approach for to peptide bond formation of peptides on the abiotic Earth, it is limited to acidic conditions. ... Compared to reactions involving hydroxy acids as catalysts, mercaptoacids catalyze thiodepsipeptide and peptide bond formation under a wider range of pH conditions, and at much milder temperatures, and at higher water activity."

First, there is no evidence these are catalysts rather than reagents. The authors should change "catalysts" to "reagents". The authors should remove "mercaptoacids catalyze".

Response: Please see the response to Reviewer 1, Comment 6, above. We removed the description of mercaptoacids as catalysts.

Second, delete "and peptide"; the products are all thiodepsipeptides.

Response: Please see the response to Reviewer 1, Comment #7 above. We modified the text to "HS-peptides" (peptides terminated with a thiol instead of an amine group) instead of "peptides".

Third, it is not possible for readers to compare the thiodepsipeptide and depsipeptide reactions as the reader doesn't know how acidic their depsipeptide synthesis must be to work. Provide the pH required to make the depsipeptide reaction work.

Response: We clarified the text. Depsipeptide formation is optimal at mildly acidic pH (pH~3.5).

Reviewer 1, Comment 9: 5) Page 3: The authors write: “It is known that thiols can condense with carboxylic acids to form thioesters.”

Where is it known? Who reported this? Under what conditions? Appropriate references are required here.

Response: The appropriate references (refs 18 and 34) were added to the revised manuscript.

Reviewer 1, Comment 10: 6) Page 4: The authors write: Thiodepsipeptides and peptides form via dry-down reactions of mercaptoacids and amino acids. To test whether thiodepsipeptides form ...”

They only observed thiodepsipeptides, they should delete “and peptides”.

Response: Please see the response to Reviewer 1, Comment #7 above. We modified the text to “HS-peptides” (peptides terminated with a thiol instead of an amine group) instead of “peptides”.

Reviewer 1, Comment 11: 7) The authors state in their response to previous comments 1 and 14: The state: “In the revised manuscript we have included data from new experiments that provide additional support for the proposed thioester-amide mechanism and that confirm the presence of amide bonds in the products of these reactions. In the first series of these experiments, the products of constant amino acid concentration (Ala) and varying concentration thiol (tg) reactions demonstrate that increasing tg increases the rate of tgA formation (which contains an amide linkage) (Now included as Supplementary Figure 1). This data shows that increasing the mercapto acid concentration increases the rate of amide bond formation, supporting the role of thiols in catalyzing amide bond formation.” And then they state: “Regarding the ratio of mercaptoacid to amino acid, we now show that the excess amino acid does not inhibit the reaction.”

The authors statements do not match the data shown.

Increased reagent (tg) correlating with increase yield of products does not provide evidence for “catalysis”. The authors should remove any reference to catalysis. They have no evidence for tg catalysis. They are over interpreting their data. What they observed is increase loading of tg gives more “thiodepsipeptide products” and increased loading of Ala gives less “products”, but as tg is in all products it is a reagent and it cannot be assumed to be a catalyst based on the data provided.

In the chromatograms for 1:1 tg:Ala and for excess Ala, in Sup. Fig 1, there is almost no product – so how can the authors say they “show that the excess amino acid does not inhibit the reaction”? The data presented shows the opposite. Ala inhibition is not necessarily a problem for publication, but the authors must comment on this in the manuscript more explicitly.

They should provide yields for reactions with 1:1 tg:Ala, and excess Ala. This is important. Without these yields how can they say Ala doesn’t inhibit the reaction? I can’t understand why the authors don’t provide a table of yields for the identifiable products for these reactions (i.e. each reaction in Sup. Fig. 1 and Sup. Fig. 2). Why only give a yield at 5:1 tg:Ala? They have developed calibration curves for the major compounds observed to form from Ala, use them. Without this quantification the reader cannot understand how well the reaction works and what the distribution of (major) products is under the different conditions reported.

Response: As suggested by the reviewer, we added product quantifications to Figures S1-S2 in the second revision (this revision) of the manuscript. Moreover, in the revised manuscript, as the reviewer requested,

we removed the description of tg as a catalyst for peptide bond formation (see response to Reviewer 1, Comment 6). As for whether Ala acts as an inhibitor of peptide bond formation, we respectfully disagree with the reviewer that this is possible. Ala alone does not polymerize (i.e. there is no product seen). This lack of reactivity is due to the high activation energy for peptide bond formation that prevents the reaction at these temperatures (e.g. Yu SS *et al.* Phys Chem Chem Phys. 2016, 18(41):28441-28450). When tg is added to the Ala solution, the rate of Ala polymerization increases (the amount of peptide bonds produced at constant time increases). These results demonstrate that Ala alone is unreactive and that tg facilitates the formation of peptide bonds between Ala. An inhibitor, by definition, is an agent that slows or interferes with a chemical reaction. But with Ala alone, there is no formation of peptide bonds.

Reviewer 1, Comment 12: 8) The authors have added two “replicates” of data that had already been presented, i.e. 5:1 tg:Ala and 2.5:1 tg:Ala in Sup. Fig 1. They appear to be different in A and B. Why?

Response: The reviewer is right that there are slight differences between the independent replicates (please note also that the y axis is shifted between panel A and panel B). These experiments were performed a year apart. The oxygen levels, albeit low, fluctuate in our chamber, slightly modulating the product distribution. Please note that when replicates are done in parallel (at the same time), the variation is minimal as shown in Supplementary figure 5.

Reviewer 1, Comment 13: 9) The peak marked “Ala” in Sup. Fig. 1B appears to be in Sup. Fig. 1A “control” (that should contain no Ala). Why? Is this contamination.

Response: No, it is not a contamination. Although they are very close in retention time, the peak in the tg control is not aligned with the Ala peak. They are roughly 10 sec apart. The tg control did not contain Ala.

Reviewer 1, Comment 14: 10) As tg is volatile the ratio of tg:Ala should decrease as the reactions are heated, and therefore lower productivity is expected over time. This should be discussed.

The authors suggest the reaction is “near equilibrium” (“These near-equilibrium reactions are driven by modest environmental changes,”). If this were true then tg loss should drive the “equilibrium” to lower yields over time, as volatile tg is lost. If this is observed the authors should report that in the paper. If it is not observed, then the reaction is not “near equilibrium” as implied, and then the authors should remove discussion of thermodynamic and equilibrium control from the paper.

How do the authors demonstrate that these reactions are “near equilibrium”?

Response:

We rely here in part on vast thiol/thioester-related literature [Thioester exchange reactions involving Cys side chains are a key element of the native chemical ligation (NCL) method of protein synthesis; for leading references, see: Dawson, P. E.; Kent, S. B. H. *Annu. Rev. Biochem.* 2000, 69, 923. Blaschke, U. K.; Silberstein, J.; Muir, T. W. *Methods Enzymol.* 2000, 328, 478-496. Hofmann, R. M.; Muir T. W. *Curr. Opin. Biotech.* 2002, 13, 297-303. For a conformationally assisted case of native chemical ligation, see: Beligere, G. S.; Dawson, P. E. *J. Am. Chem. Soc.* 1999, 121, 6332].

As described in the manuscript the net directionality of these reactions can be reversed by low energy environmental changes. That is to say that condensation products formed by drying can be hydrolyzed by increasing water activity. This facile reversal of reaction directionality is possible only for near-equilibrium systems.

It is true that tg is volatile, and some is lost under the conditions of our experiments. It is probably true that our reactions yields are decreased somewhat by tg evaporation. However, our reactions contain excess tg and therefore we still obtain significant yields.

Reviewer 1, Comment 15:11) Page 4: The authors write: “Direct infusion MS and LC-MS analyses indicated that in dry-down reactions of tg and Ala produced co-oligomers with varying compositions of tg and Ala, and with lengths of co-polymers of tg and Ala were formed (Fig. 2a). Mixtures of tg and Ala produced oligomers up to hexamers, with varying lengths and compositions of tg and Ala.”

There is no need to repeat the same information twice. Delete second instance.

Response: Deleted.

Reviewer 1, Comment 16: 12) Page 6: The authors state: “Incubation in water of a previously dried mixture of tg and Ala for 3 days at 65 °C under anoxia led to partial degradation of various oligomers, further supporting the existence of thioester bonds among the dry-down products (Supplementary Fig. 14).”

Their data in Sup. Fig 14 appears to show that tgA goes down. Why? This is not a thioester.

Response: While c(tgA) is hydrolyzing relatively quickly, as expected, it is true that there is a slight decrease in the levels of tgA. We suspect that this is due to some oxidation that occurs between the thiol groups, forming the disulfide bonds (see the increase in the tgA-S-S-tgA dimer in retention time ~14, which is labelled in the revised manuscript).

Reviewer 1, Comment 17: 13) The authors have now synthesised tgAAA and found that it cyclises to 3-methylthiazine-2,5-dione. A nice addition to this paper. But they are observing the thiopeptide undergoes a reaction that degrades the peptide bonds.

Response: It seems that our narrative was not clear. It is tgA that cyclizes to 3-methylthiazine-2,5-dione (c(tgA)). We have clarified this distinction in the text on page 6 in the first paragraph of the newly revised manuscript.

Reviewer 1, Comment 17a: 13a) The formation of tg-depsipeptides and cyclisation to form thiazine-2,5-diones must be expected to curtail the synthesis of oligomers (n>2), especially with excess tg. This reaction will degrade oligomeric products to yield “tgA dimer”.

Response: As mentioned above, tgAAA does not cyclize to c(tgA), but rather tgA cyclizes to c(tgA). We don't have evidence for back-biting of HS-peptides such as tgAAA degrading to form c(tgA) in this system.

Reviewer 1, Comment 17b: 13b) This seems to explain why the only major product is tgA, and indicates this may be a poor strategy for peptide growth beyond tgA dimer. This seems to be what the authors results demonstrate.

Response: As mentioned, we do not have evidence for back-biting of HS-peptides degrading to form c(tgA) and tgA in this system.

Reviewer 1, Comment 17c: 13c) Furthermore, on this point, the authors were asked (comment 17): What happens if “longer” oligomer products (or suspected products) are subjected to the conditions of the reaction? Are they stable? In their Response: “[They] agree with the reviewer ... Following the reviewer's suggestion, we took various products of our reactions and for simplicity purposes we focused on reactions that contain dimers...”

But, although the authors have agreed comment 17 was originally a good question they should address, they then have answered a different question (that misses the point of the original suggestion to look at longer oligomers under the reaction conditions).

They have only looked at the dimer, the shortest possible “oligomer” (assuming tgA is a “dimer” and is therefore the shortest oligomer; I don’t subscribe to this as the two parts of tgA are different, but the authors have explained already they do - so, putting the semantics of the prefix "oligo" aside), importantly the "dimer" would not be expected to display the same degradation reactivity as any longer oligomers.

Their observations with tgAAA likely vindicates the original question, and further supports the need to perform the reaction suggested in comment 17. Comment 17 specifically asked what happens when “longer” oligomers are introduced into the reaction, suspecting they would degrade, like tgAAA has been observed to degrade.

There is no doubt that the longer oligomers (and polymers) are presented as the desired products, the authors are clear they believe this is an “important stage in the origins of life”, but their results suggest longer oligo/polymers will be degraded by repeated formation of c(tgA) or c(tg)₂. It is very important to address whether the desired products (i.e. oligomers n>2) are stable to the reaction conditions.

The authors should either provide this reaction, originally requested in comment 17, to address the stability of (the desired) oligomeric products (where n>2) or make clear in the discussion within the paper that based on the results they have, they would expect products where n>2 to be disfavoured, due to peptide degradation by tg as is observed for tgAAA.

Response: We agree with the reviewer that it is possible that a longer oligomer (n>2) will backbite to “release” c(tgA) units. While such an experiment is beyond the scope of this paper, we would love to test this in the future. We agree with the reviewer that it is a possibility worth mentioning in the paper. As mentioned, we do not have evidence for backbiting of HS-peptides degrading to form c(tgA) and tgA in this system.

We added the following sentence to the Discussion:

“It is possible that oligomers produced in our reactions were reduced in length and yield by a “backbiting” mechanism, in which a terminal thiol performs an intramolecular nucleophilic attack on a thioester bond to produce a 6-membered ring c(tgA).”

Reviewer 1, Comment 18: 14) Page 6: The authors imply c(tgA) undergoes a “ring-opening polymerisation”. The authors make this sound like the predominant mode of reaction, but this is not supported by the evidence in Sup. Fig. 17.

The major product in Sup. Fig. 17 is hydrolysis, and a very small (unquantified) amount of tgAA is identified too – I see no identification of “polymers” and little evidence they might be there.

Moreover, if it is true, the product of this “polymerisation” would not be well exemplified by “tgAA” in general (as is implied in the text). The addition of A would be an initiation, the polymerisation would then go on to make tgAtgAtgAtgAA type products. In this ring-opening polymerisation the new bonds (other than the single initiation event) are thioesters, not amides. And should be expect to hydrolysis back to tgA.

Response: Supporting Figure 17 has three panels: Panel A shows a drydown reaction of c(tgA) alone, panel B shows a drydown of c(tgA) with Ala, and panel C shows a drydown of c(tgA) with tg. For the reaction shown in panel A, there is no available nucleophile for ring opening polymerization at time 0. When c(tgA) is dried-down alone, it slowly hydrolyzes, and then it is possible that the terminal thiol on tgA would ring-open c(tgA) to some extent. However, we would expect fast ring-opening polymerization of c(tgA) when it is dried-down with Ala (panel B, an amine is the nucleophile) or with tg (panel C, a thiol is the

nucleophile). Indeed, while panel A shows mostly slow hydrolysis, fast consumption of c(tgA) into longer products is observed when a nucleophile is present in the reactions from the very beginning (panels B-C).

Reviewer 1, Comment 19: 15) Page 9: Fig. 4A The authors should label the peaks they see in the starting material spectrum at 3.8 and 3.6 ppm.

Could this compound contribute to the yield of tgA? Is it a contaminant in the starting material added to the reaction?

Response: The peaks corresponding to ~3.8 ppm and ~3.6 ppm originate from the tg stock, which contains about 10% thioester-linked dimer (tg)₂. We have added these labels to the figure legend in the newly revised manuscript. As we showed, tg polymerizes readily into its polythioesters, and hence we do not think that the existence of some tg dimer is significantly affecting the kinetics of the reaction.

Reviewer 1, Comment 20: 16) The author state in their responses to the reviewer that they use imidazolium salts to adjust the pH of their reaction as it forms fluid gel-like phases upon drying: “Regarding the choice of imidazole, it was chosen due to its pKa and since it enables a gel-like state during the dry-heating process, which will enable molecule diffusion and reactivity.”

This is interesting and potentially important. The authors should communicate this to the reader in the paper.

It also opens the important question of whether the reaction does not work (or is much less effective) if another, less gel-like, buffering salt is used in place of imidazolium? Were other acids not investigated?

Further work on this might be beyond the scope of this paper - but it should be clear to the reader that all these reactions form gel-like or waxy states on drying and this is likely, or at least assumed by the authors to be, an important requirement for this reaction to proceed.

Response: We have added a description of the physical state of the dried material to Results (Page 4):

“In addition, excess mercaptoacids help maintain of a ‘gel-like’ state that enables molecule diffusion throughout the course of the reaction.”

Page 10: “The addition of imidazole to the reactions helped maintain a ‘gel-like’ state, which is likely to enable diffusion of the molecules during the reaction.”

Regarding the question about other buffering agents, we do not have an example for a reaction in which pH was adjusted but the gel-like state was not maintained. As mentioned in our first revision, we also carried out reactions using NaOH and showed that some reactivity is observed at basic conditions, although the highest reactivity is observed at acidic pH.

Reviewer #2:

The manuscript by Frenkel-Pinter et al has been improved by the review process. The realization of thiopeptides through dry down experiments is a natural extension of previous work showing peptide formation, and the occurrence at elevated pH - and also in water - provides the community with a significant advance in our understanding of how peptides can be formed under conditions relevant to the early Earth.

Response: We thank the reviewer for their positive assessment of our work and for constructive suggestions, which have improved our manuscript.

I have a few small final comments that I hope might be useful to the authors.

Reviewer 2, Comment 1: While re-reading the statements in the abstract and introduction which state that previous work required “activated building blocks or condensing agents” I recalled Huber et alia’s work from 1998 and 2003:

31 JULY 1998 VOL 281 SCIENCE

15 AUGUST 2003 VOL 301 SCIENCE.

In those papers, peptides were formed in a reaction with CO.

Related to this, Leman et al demonstrated high yields of peptides synthesized with COS. SCIENCE VOL 306 8 OCTOBER 2004

Synlett 2017, 28, 68–72

These results showed at least 2 paths for making peptides in water without “activated building blocks” and while CO and COS might be viewed as a “condensing agent” these are certainly relevant to Archean atmospheres:

<https://www.pnas.org/content/106/35/14784>

To this reviewer, softening the framing of the current work in light of these past observations seems prudent and it would be useful for readers if at least one of the Huber and Leman papers were cited.

Response: We thank the reviewer for these suggestions. We added the references to the paper in the introduction. We also modified our statement in the introduction and abstract from “all abiotic reactions reported thus far for peptide bond formation via thioester intermediates have relied on activated building blocks or condensing agents, which are of questionable prebiotic relevance” to “all abiotic reactions reported thus far for peptide bond formation via thioester intermediates have relied on high energy molecules, which usually suffer from short half-life in aqueous conditions”. We also added the following description in the Introduction:

“Leman and co-workers showed that carbonyl sulfide can facilitate peptide bond formation via generation of α -amino acid thiocarbamate, which is cyclized into the highly reactive *N*-carboxyanhydride.”

Reviewer 2, Comment 2: Also about the references: “thioester intermediates enable the synthesis and degradation of peptides, fatty acids, sterols, and porphyrins.7,8 “Are references 7 and 8 correct here? They don’t seem to discuss what is being referred to Perhaps an article or book chapter like this would serve the

authors intent (though there are of course others) 7.11 - Coenzyme A Biosynthesis and Enzymology Erick Strauss

<https://doi.org/10.1016/B978-008045382-8.00141-6> and: “ long before the great oxidation event that occurred only around 2 billion years ago.45” Reference 45 may be out of place here. A more appropriate reference would be a recent and excellent review: Catling and Zahnle, *Sci. Adv.* 2020;6:eaax1420

Response: We thank the reviewer for these helpful suggestions. We agree with the reviewer that these references are more appropriate. We have decided to replace the previously used references with the ones suggested by the reviewer.

Reviewer 2, Comment 3: Also please change “around 2 billion years ago” to “around 2.4 Ga”. 400 million years is a long time.

Response: We modified the text to “around 2.4 Ga”, as suggested by the Reviewer.

Reviewer 2, Comment 4: Supplemental figure 9 and the formation of tgAAA. These data are very unconvincing and could probably be used to argue that tgAAA was not synthesized in the reaction if I'm understanding it correctly. In panel A, the dry down does not show the expected peak. In panel B) the spike adds only a very small peak to the peak labeled with an arrow. Why is it so small? Can the authors verify this?

The authors write that this was 50 μ M of tgAAA; what was the spike amount in panel A?

Response: We understand the source of confusion regarding tgAAA and have modified the manuscript to clarify. In panel A we injected 10 mM tgAAA (black trace), whereas in panel B indeed only 50 μ M (final concentration) were spiked into a dried tg+Ala reaction. In the revision we indicate that the difference in the injected tgAAA concentration between the two panels by stating the exact concentration of the standard in panel A as well. We thank the reviewer for bringing this to our attention. The reviewer is right that the amount of tgAAA formed is small. Based on our calculation of the calibration curves (Supplementary Figure 11) we quantified that 0.17% of Ala was converted into tgAAA. While this is a low yield, it allows us to state that the product does form to some extent. In that context we would also like to note that our reactions were only carried out for 1 week and perhaps longer reaction times will lead to higher yield of products such as tgAAA.

Reviewer 2, Comment 5: In all of the spiked and standard sample chromatograms shown in the paper, it would be beneficial to indicate the concentration - or better yet - the mol amount injected.

Response: Done.

REVIEWERS' COMMENTS

Reviewer #1 (Remarks to the Author):

All reviewer comments have been addressed.

Reviewer #2 (Remarks to the Author):

In this second revision, the manuscript has been again improved in terms of clarity (labeling figures, presentation, etc). The main findings of a new route to peptide bond formation are exciting and relevant for a number of research areas.

While all of my comments have been addressed, I may add as a note from reading those of reviewer 1, who raises an interesting but still to be unaddressed point about the tg:ala ratios and their effects on product yields. In the rebuttal, the authors reply that Ala cannot be an inhibitor because it does not react on its own, but this logic does not help in the opinion of this reviewer. Clearly, adding more of one type of a substrate is changing the product distributions; whether that is an "inhibition" or if it is favoring the operation of a different chemical pathway (or halting another), is yet unknown. Thus I suggest the author's discussion as it stands is reasonable and sufficient until further work is done; that the product yields are sensitive to initial ratios an interesting and supplemental finding to the current work.